

# Derivation of horizontal and vertical wavelengths using a scanning OH(3-1) airglow spectrometer

Sabine Wüst [1*], Thomas Offenwanger [1], Carsten Schmidt[1], Michael Bittner [1,2], Christoph Jacobi [3], Gunter Stober [4], Jeng-Hwa Yee [5] , Martin G. Mlynczak [6], James M. Russell III [7]

[1]Deutsches Zentrum für Luft- und Raumfahrt, Deutsches Fernerkundungsdatenzentrum, 82234 Oberpfaffenhofen, Germany
[2]Universität Augsburg, Institut für Physik, Augsburg, Germany
[3]Universität Leipzig, Institut für Meteorologie, Leipzig, Germany
[4]Institut für Atmosphärenphysik, Kühlungsborn, Germany
[5]Applied Physics Laboratory, The Johns Hopkins University, Laurel, USA
[6]NASA Langley Research Center, Hampton, USA
[7]Center for Atmospheric Sciences, Hampton, USA

*Correspondence to*: Sabine Wüst (sabine.wuest@dlr.de)

## Abstract

For the first time, we present an approach to derive zonal, meridional and vertical wavelengths as well as periods of gravity

waves based on only one OH* spectrometer addressing one vibrational-rotational transition. Knowledge of these parameters is a precondition for the calculation of further information such as the wave group velocity vector.

OH(3-1) spectrometer measurements allow the analysis of gravity wave periods, but spatial information cannot necessarily be deduced. We use a scanning spectrometer and the harmonic analysis to derive horizontal wavelengths at the mesopause above Oberpfaffenhofen (48.09°N, 11.28°E), Germany for 22 nights in 2015. Based on the approximation of the dispersion

relation for gravity waves of low and medium frequency and additional horizontal wind information, we calculate vertical wavelengths afterwards. The mesopause wind measurements nearest to Oberpfaffenhofen are conducted at Collm (51.30°N, 13.02°E), Germany, ca. 380 km northeast of Oberfpaffenhofen by a meteor radar.

In order to check our results, vertical temperature profiles of TIMED-SABER (Thermosphere Ionosphere Mesosphere Energetics Dynamics, Sounding of the Atmosphere using Broadband Emission Radiometry) overpasses are analysed with

respect to the dominating vertical wavelength.





## 1 Introduction

In order to analyse atmospheric motions like gravity waves, the upper mesosphere/lower thermosphere is addressed by a variety of measurement techniques: airglow spectroscopy and imaging as well as lidar systems are probably the most prominent ones in the Network for the Detection of Mesospheric Change (NDMC, https://www.wdc.dlr.de/ndmc).

Depending on the instrument and the retrieval, different techniques are sensitive for different wave parameters. While lidar measurements, for example, allow the derivation of vertical wavelengths (see, e.g., Rauthe et al., 2006, 2008; Mzé et al., 2014), horizontal wavelengths of larger-scale gravity waves can be investigated by meteor radars (Oleynikov et al., 2005, 2007). Airglow imaging can be used for the analysis of horizontal wavelengths of shorter-scale waves. A sufficient number of measurements in a specific time period provided, also wave periods can be derived by these techniques (for instance,

Garcia et al., 1997; Taylor et al., 2003; Yamashita et al., 2009). OH-airglow spectroscopy however (e.g., Bittner et al., 2000; Mulligan et al., 1995), if it is based on only one vibrational OH* transition, can exclusively deliver wave periods. Wachter et al. (2015) show that the combination of three airglow spectrometers measuring under different azimuth angles allows the additional derivation of horizontal wavelengths. Due to the setup of the three instruments, their fields of view (FoV) and the data analysis technique, the retrieved wavelengths lie mostly in the range of some 100 km, the addressed wave periods range

from 1 to 14 h with a maximum between 2 and 4 h. Small-scale horizontal features in the order of some 10 km or even turbulent structures like they are observed with OH* cameras as shown by Sedlak et al. (2016) and Hannawald et al. (2016) cannot be investigated based on this approach.

Schmidt et al. (2017) introduce a method to additionally derive vertical wavelengths from OH* spectrometer measurements by observing two vibrational transitions, OH(3-1) and OH(4-2). Following the work of von Savigny (2012), the radiation

emitted by the different vibrational transitions originates from slightly different heights which are separated by a few 100 m. For approximately 40% of the wave events, a vertical wavelength can be derived which lies in the range of 5–40 km. Of course, the same approach can be applied to measurements of different airglow species peaking at different heights, for example, OH(6-2) and $O_2b$(0-1) which are separated by ca. 7 km.

Here, for the first time, we present an approach to derive zonal, meridional and vertical wavelengths as well as wave periods

based on only one OH* spectrometer addressing one vibrational-rotational transition. Knowledge of these parameters is a precondition for the calculation of further information like the wave group velocity vector or the vertical flux of horizontal wave pseudomomentum, for example (see e.g. Fritts and Alexander, 2003). However, the derivation of these values is beyond the scope of this manuscript.

For low- and medium-frequency waves, the wave vector is related to the intrinsic wave frequency via the Brunt-Väisälä

frequency and the Coriolis parameter (dispersion equation, see equation (38) in Fritts and Alexander, 2003). Based on the work of Wachter et al. (2015), we construct a scanning OH* spectrometer (section 2.1) which allows the derivation of periods and zonal as well as meridional wavelengths (method: section 3.1, and results: section 4.1). We then use literature values of the Brunt-Väisälä frequency (see e.g. Wüst et al., 2016, 2017b) and the nearest mesopause wind measurements





which are performed by a meteor radar (section 2.3) in order to estimate the vertical wavelengths (method: sections 3.1, and results: section 4.2). The scanning spectrometer measures at Oberpfaffenhofen (48.09°N, 11.28°E), Germany, the meteor radar is deployed at Collm (51.30°N, 13.02°E), Germany, ca. 380 km northeast of Oberpfaffenhofen; specific focus is therefore put on a thorough uncertainty estimation (section 3.2). Finally, the results are compared to vertical wavelengths

5  extracted from collocated TIMED-SABER temperature profiles (section 2.2, and 4.2).



## 2 Measurements and data

### 2.1 Infrared spectrometer GRIPS

The nightly airglow observations presented here are performed with the scanning infrared spectrometer GRIPS 14 (GRound based Infrared P-branch Spectrometer) at Oberpfaffenhofen (48.09°N, 11.28°E), Germany, July–November 2015. The

instrument operates in the spectral range of 1.5 µm to 1.6 µm. Therefore, observations are only possible under (nearly) cloudless conditions. They address a height of ca. 86 km (e.g. Wüst et al., 2016; Wüst et al, 2017b).

Concerning its basic components and its data processing, GRIPS 14 is identical to the GRIPS instruments described by Schmidt et al. (2013). The technical layout of the scanning mirror is designed to result in three FoV forming an equilateral triangle (zenith angles: 30°) in the mesopause region with the fourth FoV being in the center of the triangle, in zenith

direction. The edge length of the FoV triangle amounts to 90 km. Due to the finite aperture of the GRIPS 14, the FoV sizes of the triangle are approximately 880 km², while the zenith FoV is smaller with ca. 560 km². The instrument acquires spectra with a temporal resolution of 15 s. Thus, it is possible to get airglow spectra from four FoV in approximately one minute.

The rotational temperature derived from an individual spectrum can typically exhibit an uncertainty of ±8 K. In order to improve the signal-to-noise ratio for the intended analysis five minute mean values are calculated for each FoV. Additional

care has been taken to ensure that the data quality of each FoV is comparable to the others by manually inspecting each night: due to the geographic location of Oberpfaffenhofen just north of the Alps, it frequently happens that clouds form predominantly in the southern FoV or that the moon passes through just one of the FoV. These cases are excluded from further analysis.

### 2.2 TIMED-SABER

On 7th December 2001, the TIMED satellite was launched. Soon, the on-board limb-sounder SABER started to deliver vertical profiles of kinetic temperature on a routine basis. The profiles cover the height range from approximately 10 km to more than 100 km. The vertical resolution is ca. 2 km (Mertens et al., 2004; Mlynczak, 1997) which is suitable for the investigation of gravity wave activity. On a given day, the latitudinal coverage extends from about 52° latitude in one hemisphere to 83° in the other (Russell et al., 1999). This viewing geometry alternates once every 60 days due to 180° yaw

manoeuvers of the TIMED satellite (Russell et al., 1999). In total, ca. 1200 temperature profiles are available per day. An overview of the large number of SABER publications is available at http://saber.gats-inc.com/publications.php.

Measurements of infrared emission from carbon dioxide in the 15 um spectral interval are used in the SABER temperature retrieval. It is based on a comprehensive forward radiance model incorporating dozens of vibration-rotation bands of $CO_2$, including isotopic and hot bands, and solving the full set of coupled radiative transfer equations under non-LTE, i.e., under

conditions that depart from Local Thermodynamic Equilibrium. From the temperature retrieval version 1.03 on, NLTE algorithms for kinetic temperature were employed (López-Puertas et al., 2004; Mertens et al., 2004, 2008). This is certainly



one of the main challenges for $CO_2$ based temperature retrievals in the mesosphere and upper levels. Comparisons with reference data sets generally confirm good quality of SABER temperatures (Remsberg et al., 2008).

We use TIMED-SABER temperature data between 45.4°N and 50.8°N and 8.6°E and 14.0°E (ca. 300 km distance from Oberpfaffenhofen (48.09°N, 11.28°N)) in its latest version (2.0). It was downloaded from the SABER homepage (saber.gats-inc.com).

The data were detrended between 100 km and their height minimum using an iterative cubic spline approach as it is described in Wüst et al. (2017a) with a distance of 10 km between two spline sampling points. This results in a maximal detectable wavelength of 20 km (in the detrended data series). We restrict further analysis to a relatively small height interval of 60—80 km which is just below the height range addressed by GRIPS. This is due to the following reasons. Especially during summer (May–August), a time period which is also covered in this study, the mesopause is low and reaches ca. 86 km ± 3 km (von Zahn et al., 1996; She et al., 2000). Sharply changing temperature gradients are always a challenge for a de-trending procedure and artificial signals in the detrended data cannot be excluded here. This is the reason why we investigate only heights below 80 km with the harmonic analysis. The majority of commonly-used spectral analysis techniques like the fast fourier transform, the maximum entropy method and also the harmonic analysis approach, all assume the waves are stationary and therefore a constant wave amplitude. Alternative analyses suited for non-stationary time series like, for example, the wavelet analysis often suffer from a relatively coarse spectral resolution. Therefore, we restrict our analysis to the smallest possible height interval which is equal to the maximal wavelength detectable in the detrended data series.

## 2.3 Meteor wind radar

The VHF SKiYMET meteor radar located at Collm has been operated nearly continuously since July 2004 (Jacobi et al., 2007, 2009). It measures winds, temperatures, and some meteor parameters at altitudes between approximately 80 and 100 km. The radar uses the Doppler shift of the reflected radio wave from ionized meteor trails to obtain radial velocities along the line of sight of the radio wave.

The radar operates at a frequency of 36.2 MHz, with 15 kW peak power at a pulse repetition frequency of 625 Hz. The transmit antenna is a crossed dipole one, while the 5 receiving antennas during 2015 were 2 element Yagi antennas, forming an interferometer to detect the meteor position. The radar delivers hourly mean horizontal wind values through projection of the horizontal hourly wind components to the individual radial winds under the assumption that vertical winds are small. The procedure is described in Hocking et al. (2001). We used height gates of 3 km width for the fit. A more recent version of the wind fitting technique and error estimation of meteor radar winds can be found in Stober et al. (2017).

In order to estimate the error that arises from using the Collm observations for the wind field over Oberpfaffenhofen at a distance of about 380 km, we evaluated the differences of winds measured by the Collm radar and the 53.5 MHz OSWIN VHF radar (Latteck et al., 1999) at Kühlungsborn (54.1°N, 11.8°E), about 330 km distance from Collm, during a half-year campaign in 2004/05 (Viehweg, 2006). The OSWIN radar had been operated as a meteor radar (Singer at al., 2003), with the same analysis procedure than applied at Collm. The Collm-Kühlungsborn differences were increasing from -0.7 ± 22.3 m/s





at 85 km to -2.5 ± 25.5 m/s at 94 km for the zonal component, and -0.1 ± 20.3 m/s at 85 km to -2.05 ± 24.9 m/s at 94 km for the meridional component. The small biases may be explained by the mean northward gradients of the horizontal winds, which in winter at these heights are positive for both the zonal and meridional wind component. The standard deviation is owing to waves, turbulence, and uncertainties of both systems.

5    Therefore, when using Collm data for estimating winds over Oberpfaffenhofen, the standard deviation of about 20 m/s may be considered as a good guess for the dynamical induced wind differences.





## 3 Analysis methods

### 3.1 Derivation of 3D wave vector

The basic idea of the algorithm applied here for the calculation of horizontal wavelengths from a scanning GRIPS instrument is already mentioned in Wachter et al. (2015). In contrast to their publication, we derive OH-temperatures for four instead of

three FoV with one scanning GRIPS instrument instead of three individual (non-scanning) ones. Since three FoV are sufficient for the calculation of horizontal wavelengths, we use the additional information for the estimation of uncertainty intervals.

We apply the harmonic analysis (all-step mode, see for example Bittner et al. (1994) or Wüst and Bittner (2006)) to the four

nightly time series and search for four identical periods. Further analysis steps are restricted to results which are characterized by a period longer (shorter) than 60 min (the measurement time) and an amplitude larger than or equal to 1 K. This is in accordance with the approach and the results of Wachter et al. (2015) (see their section 2.2).

Since four different triangles can be derived from four different FoV, we apply the algorithm described in Wachter et al. (2015) to each possible triangle combination. So, we get information about the horizontal wavelengths $\lambda_h$ (wave numbers $k_h$)

from each of the four triangle combinations for four waves at maximum. Zonal and meridional wavelengths (numbers) $\lambda_x$ ($k$) and $\lambda_y$ ($l$), phase velocities, and propagation directions can be derived. The mean parameters are calculated for each wave, and the mean absolute difference between the individual values and the mean parameters are taken as a measure of uncertainty.

Since phase velocities reported in literature do in most cases not exceed 150 m/s (e.g., Nakamura et al., 1999; Taylor et al.,

2009; Tang et al., 2014; Wachter et al., 2015), only waves with a mean phase velocity of 150 m/s at maximum and a mean horizontal wavelength of less than or equal to 3600 km are subject of further analysis steps. Additionally, a maximal difference of 90° between the four different wave vectors is accepted. It turned out that this criterion is the strictest one: if it is fulfilled, the others are met as well.

Going one step further than Wachter et al. (2015), we then use the dispersion relation for the estimation of vertical wavelengths. According to linear theory (see, for example, Fritts and Alexander, 2003), it holds:

$$m^2 = \frac{(k^2 + l^2)(N^2 - \sigma^2)}{(\sigma^2 - f^2)} - \frac{1}{4H^2} \tag{1}$$

where

$m$ is the vertical wave number,

$N$ is the Brunt-Väisälä frequency,

$\sigma = \omega - k\bar{u} - l\bar{v}$ is the intrinsic frequency (the frequency that would be observed in a frame of reference moving with the background wind $(\bar{u}, \bar{v})$),





$\omega$ is the frequency derived by the harmonic analysis,

$f = 2 \cdot \frac{2\pi}{86164\,s} \cdot \sin\beta$ is the Coriolis parameter with respect to the latitude $\beta$, which reaches typically $10^{-4}\,\text{s}^{-1}$ for mid-latitudes, and

$H$ is the scale height.

For medium- and low frequency waves ($\sigma \sim f$ or $N \gg \sigma \gg f$) the dispersion relation simplifies to

$$\lambda_z = \frac{2 \cdot \pi \cdot \sqrt{\sigma^2 - f^2}}{N \cdot k_h} = \frac{2 \cdot \pi \cdot \sqrt{(\omega - k \cdot u - l \cdot v)^2 - f^2}}{N \cdot k_h} \qquad (2)$$

where $k_h$ is the horizontal wave number (see formula (38) in Fritts and Alexander, 2003). Due to the selection criteria for frequency and horizontal wave numbers, $\omega$, $k$, and $l$ are rather small and the use of this approximation is justifiable.

10   Information about mesopause wind velocities above Oberpfaffenhofen is not available. Since tides, which are variable from day to day, play an important role in this height range, we do not rely on climatological wind values but make use of wind measurements performed with the wind meteor radar at Collm in order to estimate the intrinsic frequency.

Since GRIPS only measures the temperature at about 86 km height, but not the temperature gradient, the values for the Brunt-Väisälä (angular) frequency are taken from Wüst et al. (2016) (see table 1). They are calculated on a monthly base and

15   rely on a comparison between CIRA-86 (COSPAR International Reference Atmosphere, Committee on Space Research and NASA National Space Science Data Center, 2006) and TIMED-SABER values.

### 3.2 Error estimation

Since $\omega$ is calculated using four different time series and applying a variety of quality criteria, we argue that the error of $\omega$ is

20   negligible. Following error propagation, the error of $\lambda_z$ then sums up to

$$\Delta\lambda_z = \sqrt{\left(\frac{\partial\lambda_z}{\partial k}\Delta k\right)^2 + \left(\frac{\partial\lambda_z}{\partial l}\Delta l\right)^2 + \left(\frac{\partial\lambda_z}{\partial N}\Delta N\right)^2 + \left(\frac{\partial\lambda_z}{\partial u}\Delta u\right)^2 + \left(\frac{\partial\lambda_z}{\partial v}\Delta v\right)^2} \qquad (3)$$

with

$$\frac{\partial\lambda_z}{\partial k}\Delta k = -\lambda_z \cdot \Delta k \cdot \left(\frac{\sigma \cdot u}{\sigma^2 - f^2} + \frac{k}{k^2 + l^2}\right) \qquad (4)$$

$$\frac{\partial\lambda_z}{\partial l}\Delta l = -\lambda_z \cdot \Delta l \cdot \left(\frac{\sigma \cdot v}{\sigma^2 - f^2} + \frac{l}{k^2 + l^2}\right) \qquad (5)$$

$$\frac{\partial\lambda_z}{\partial N}\Delta N = -\lambda_z \cdot \frac{\Delta N}{N} \qquad (6)$$

$$\frac{\partial\lambda_z}{\partial u}\Delta u = -\lambda_z \cdot \frac{\sigma \cdot k}{\sigma^2 - f^2} \cdot \Delta u \qquad (7)$$





$$\frac{\partial \lambda_z}{\partial v} \Delta v = -\lambda_z \cdot \frac{\sigma \cdot l}{\sigma^2 - f^2} \cdot \Delta v \tag{8}$$

Following Wüst et al. (2016) and Wüst et al. (2017b), $\frac{\Delta N}{N}$ is ca. 10%, $\Delta u$ and $\Delta v$ are ca. 20 m/s (see section 2.3), and $\Delta k$ and $\Delta l$ are estimated as stated above (see section 3.1). Now, $\Delta \lambda_z$ can be calculated.



## 4 Results and discussion

### 4.1 Horizontal wavelengths

Due to the rather strict quality criteria, horizontal wavelengths for 31 wave events in only 22 nights can be identified during the measurement period. For the majority of cases, the horizontal wavelength is shorter than 1000 km, with a maximum of
the distribution between 600 and 800 km (fig. 1a). The phase velocity reaches 140 m/s at maximum and ranges mostly between 20 and 40 m/s (fig. 1b). A preferred propagation direction cannot easily be identified (fig. 1c). The data cover the time period from July to November; since the propagation direction is supposed to show seasonal variations (see Wachter et al. (2015), and references therein) and our data base is rather small, a conclusive picture cannot be drawn here.

The values for the different parameters agree well with literature. Phase velocities up to 80–100 m/s are reported, for
example, by Nakamura et al. (1999), Suzuki et al. (2004), and Taylor et al. (2009). Tang et al. (2014) and Wachter et al. (2015), e.g., find horizontal phase speeds of up to 160–180 m/s. The horizontal wavelengths cannot easily be compared since many authors focus on smaller horizontal scales (see, for example, Tang et al., 2014; Taylor et al., 2009; Hannawald et al., 2016; Sedlak et al., 2016). However, Reid (1986) presents in his fig. 6 a good overview of horizontal wavelengths measured by different techniques at various locations between 60 and 100 km height. Here, it becomes clear that horizontal
wavelengths of the order of $10^3$ km were already observed in earlier studies.

### 4.2 Vertical wavelengths

Since TIMED-SABER temperature profiles are used for validating the vertical wavelengths derived from the scanning GRIPS, we identify all nights with co-located TIMED-SABER measurements around Oberpfaffenhofen. Depending on the orbit of TIMED, it happens that not exactly one vertical temperature profile is suitable but none or more than one. Since the
wind velocity changes considerably during night, we calculate linearly weighted wind speeds from the hourly means of the wind data according to the overflight time of TIMED. In one case, the respective hourly-averaged wind data do not exist. Therefore, 19 horizontal wavelengths of the 31 mentioned in section 4.1 referring to 14 of 22 nights can be used for further analysis. The data availability of TIMED-SABER and meteor wind measurements allows the calculation of 48 vertical wavelengths (see table 2).

In three cases, the wavelengths are shorter than 2 km (no. 23, 30 and 32 in table 2). This does not seem to be a realistic value for a layer with a full width at half maximum of ca. 8–9 km (see fig. 9 in Wüst et al. (2016); Wüst et al. (2017b)). Furthermore, as Trinh et al. (2015) show in their fig. 7 a, SABER is not sensitive for vertical wavelengths shorter than ca. 2.5 km. In four cases (two times two nearly simultaneously-measured SABER profiles), the wavelengths are rather long with
ca. 38.0 km (no. 36 and 37 in table 2) and 45.9 km (no. 44 and 45 in table 2). This is in principle possible and was already observed in the past (see, for example, Manson, 1990, and Stober et al., 2013) but hard to verify here since we use SABER profiles only between 60 and 80 km—see section 2.2 for an explanation. In two cases (two nearly simultaneously-measured



SABER profiles), the result is imaginary (no. 42 and 43 in table 2), which means that the wave cannot propagate vertically and is damped. We cannot verify this case either. So, 39 cases (81% of 48 vertical wavelengths) show reasonable and verifiable results.

The mean vertical wavelength derived by the combination of GRIPS and the meteor radar is 12.5 km (fig. 2a); this is approximately equal to the scale height of 12.75 km (after CIRA-86) at ca. 87 km height and ensures the applicability of linear wave theory. A mean vertical wavelength of ca. 12.5 km is a reasonable result: Rauthe et al. (2008), for example, investigate vertical lidar temperature profiles between 1 and 105 km height recorded at Kühlungsborn (54.1°N, 11.8°E), which is located about 800 km north of Oberpfaffenhofen, with a wavelet analysis. In their fig. 5 b, they show the

dominating vertical wavelength depending on month. For a maximum height of 80 km, it ranges between 13 and 15 km. Senft et al. (1991) report a similar finding for Urbana (40.1°N, 88.2°W), United States of America. Based on 60 nights of Na-lidar measurements, they find that characteristic vertical wavelengths vary between 8.9 and 27 km. The annual mean reaches 14.1 km, if one refers only to summer values, it is 12.7 km, winter values show a mean of 15.5 km.
The mean error following equation (3) sums up to 59% (fig. 2b). In nearly all cases, the largest contribution to the individual

$\Delta\lambda_z$ is due to the wind uncertainty (formula (7) and (8)).

In order to compare the vertical wavelengths derived by GRIPS with SABER measurements, the individual error is now added and subtracted from the respective vertical wavelength. Then, the harmonic analysis is used for searching the detrended TIMED-SABER temperature profiles for one vertical wavelength in this interval. The sensitivity of TIMED-

SABER depends on the vertical and horizontal wavelength along the line of sight. For a wave with 12.5 km vertical wavelength, the horizontal wavelength (along the line of sight) needs to be 450 km at least, to ensure that SABER captures it with 50% and more of its original amplitude (see Trinh et al., 2015, e.g.). The horizontal wavelength along the line of sight is always larger or equal to the true horizontal wavelength. The TIMED-SABER data are therefore suitable for our purpose (compare fig. 1 a).

The oscillations identified by the harmonic analysis explain ca. 57% of the TIMED-SABER temperature variability on average which can be judged as a good value (fig. 3). The mean individual difference between the vertical wavelengths of both data sets reaches ca. 4.5 km or 41% relative to the GRIPS wavelength (fig. 4a and b). A mean error of 59% as calculated based on error propagation can be therefore regarded as a conservative estimate.

We conclude that the presented approach provides reasonable results for the 3D wave vector which agree with the results based on independent measurements. However, the data base is not very large. Like other measurement techniques or approaches, also this one is sensitive to certain horizontal and vertical wavelengths: the vertical extension of the OH*-layer limits the sensitivity of the GRIPS-instrument for vertical wavelengths to a few kilometres at least (see Wüst et al., 2016 for a comprehensive overview). The sensitivity for horizontal wavelengths is determined by the distance between the different



FoV (ca. 90 km here) and their sizes (see Wüst et al. (2016) for an estimation of this effect), as well as by the quality of the data, which strongly depends on the weather: only, if a phase difference unequal to zero for the individual time series can be identified, the derivation of the horizontal wave vector and in the following of the vertical wave number is possible.



## 5 Summary

Using a scanning OH-spectrometer at Oberpfaffenhofen (48.09°N, 11.28°E), Germany, we derive periods and horizontal wavelengths at the mesopause which are typical for gravity waves (ca. 1–10 h, some 100 km or 1000 km). Based on the dispersion relation, additional horizontal wind information allows the calculation of vertical wavelengths. The nearest

mesopause wind measurements are carried out at Collm (51.30°N, 13.02°E), Germany, ca. 380 km northeast of Oberpfaffenhofen by a meteor radar. We assume that these values are also valid for Oberpfaffenhofen within an uncertainty of ±20 m/s.

Ca. 80% of the vertical wavelengths range between 5 km and 19 km. These values appear reasonable compared to literature and taking into account the vertical extension of the OH-layer. In three cases (ca. 6%), the values are not plausible.

The results are compared to vertical wavelengths derived from collocated detrended TIMED-SABER measurements. Although the spectrometer and the meteor radar are deployed ca. 380 km apart from each other, the vertical wavelengths based on the spectrometer-radar data combination and the satellite data only show a mean difference of 4.5 km or 41% (relative to the GRIPS wavelength). We conclude that the presented combination of measurements provides a good estimate of the vertical wavelengths on average.

**Acknowledgement**

We thank the Bavarian State Ministry of the Environment and Consumer Protection (BayStMinUV, VAO-project LUDWIG, TP I/3, project number TUS01 UFS-67093) and the German Ministry for Education and Research (BMBF, Grant agreement No: 01LG1206A) for funding.

Furthermore, we thank Ricarda Linz, formerly at DLR, for detrending the SABER data and Paul Wachter, DLR, for a
preliminary setup of the scanning GRIPS-instrument.

Processing and long-term archiving of the data is provided by the World Data Center for Remote Sensing of the Atmosphere (WDC-RSAT, http://wdc.dlr.de). The measurements are part of the Network for the Detection of Mesospheric Change, NDMC (https://www.wdc.dlr.de/ndmc).



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



**(a)**

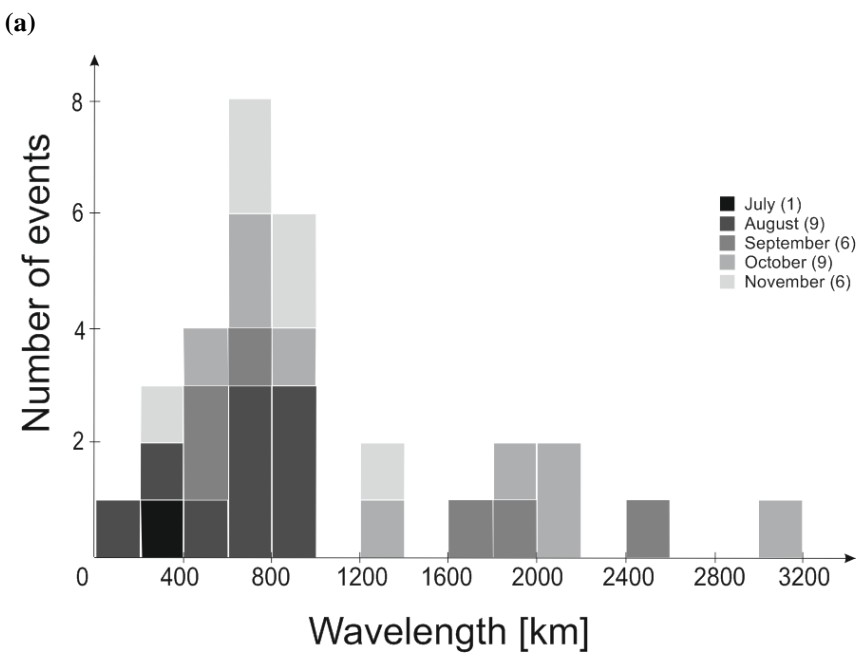

**(b)**

**(c)**

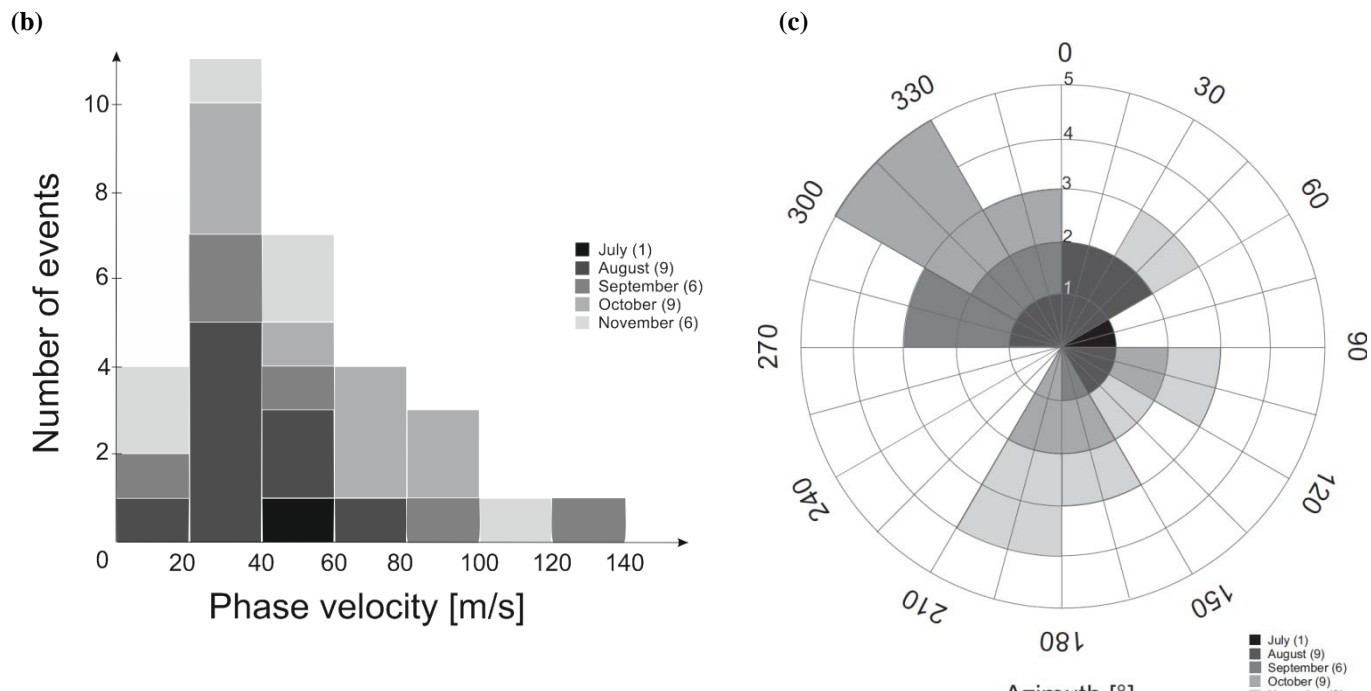



**Figure 1: The histogram of the horizontal wavelengths (a) shows that the majority of detected wave events reaches up to 1000 km. The distribution of phase velocities (b) is smoother and has a maximum between 20 and 40 m/s. The diagram of horizontal propagation directions (c) does not show a conclusive picture. The different grey colours refer to the different months.**





**(a)**

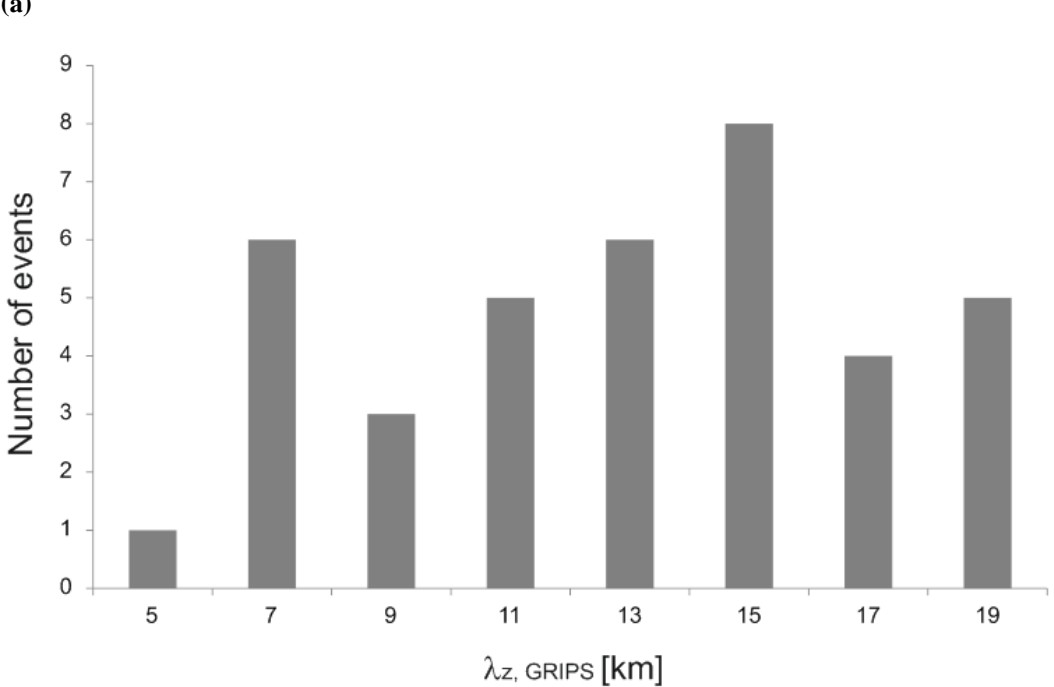

**(b)**

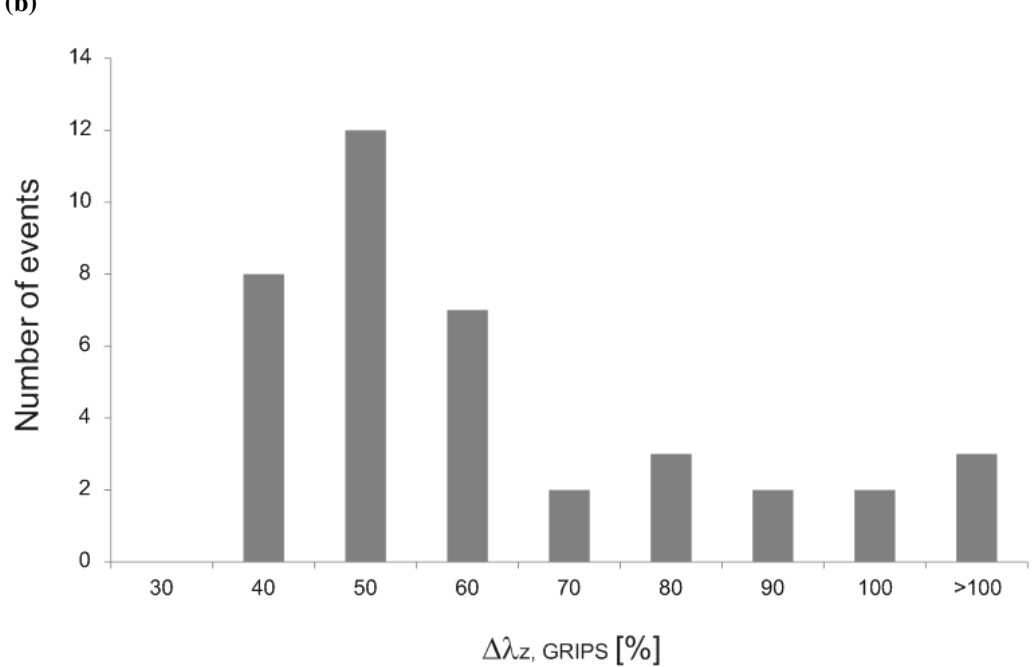

5   **Figure 2 The mean calculated vertical wavelength is ca. 12.5 km, while the individual values spread from 5–19 km (a). The mean error of the vertical wavelength $\Delta\lambda_z$ based on error propagation calculations reaches ca. 59% (b). It does not fall under 40%, in three cases, it is more than 100%.**



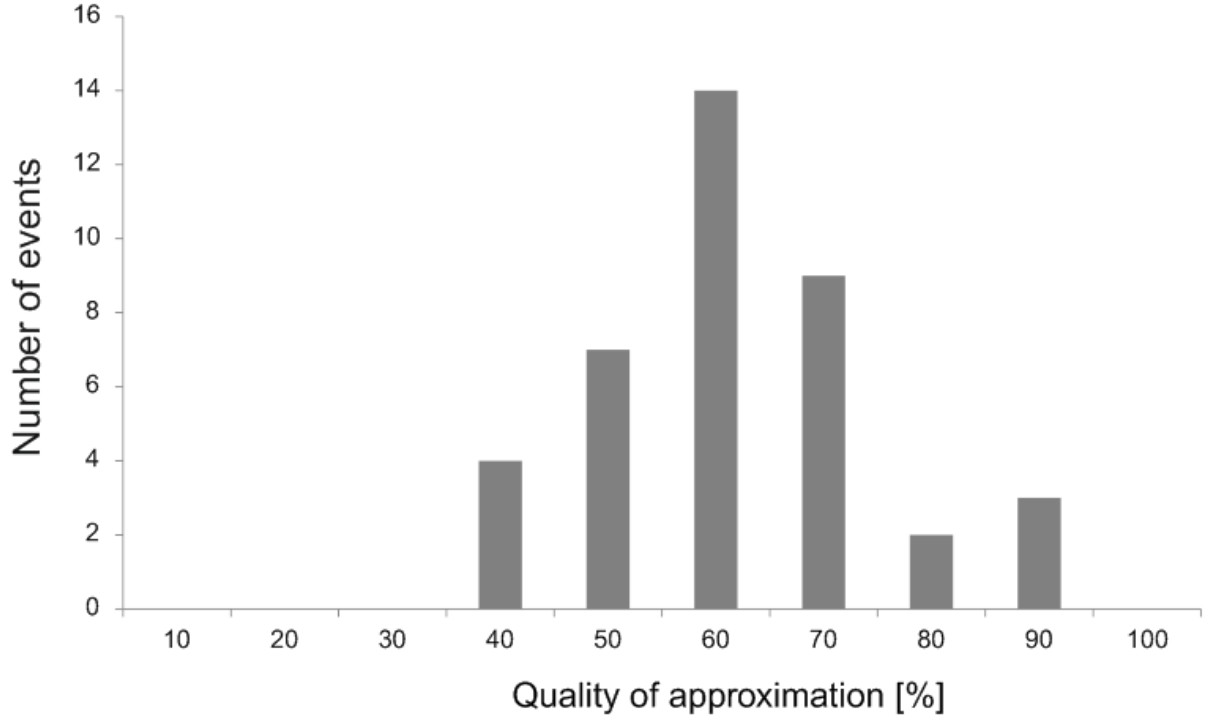

**Figure 3 The parameter "quality of approximation" describes how well the oscillation identified by the harmonic analysis describes the data variability. On average, one oscillation can explain 57% of the data variability; it ranges between 40% and 90%.**





**(a)**

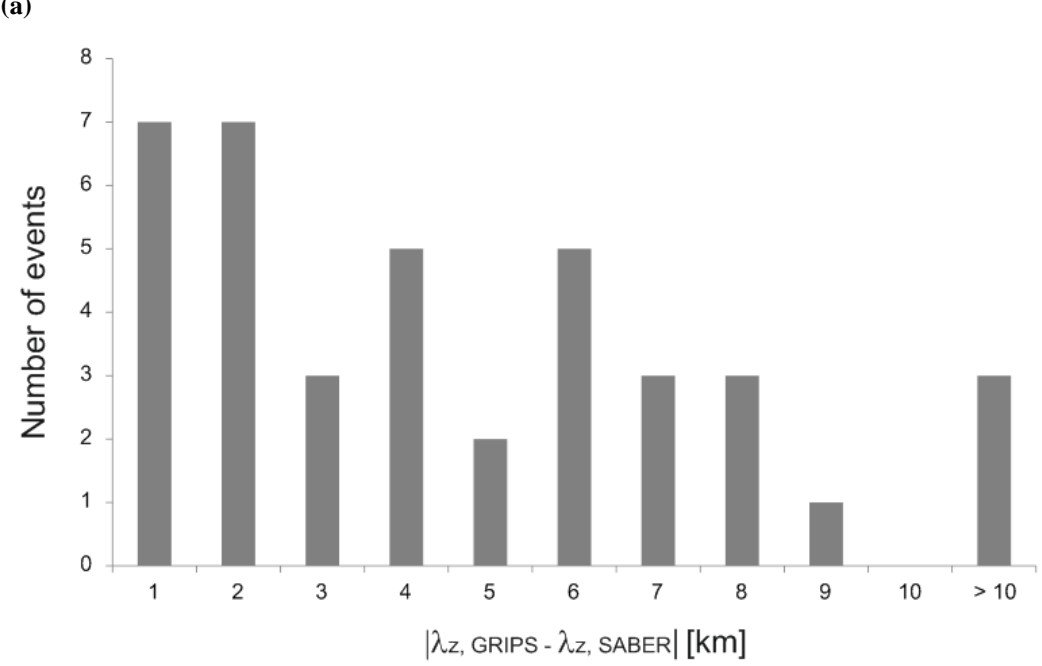

**(b)**

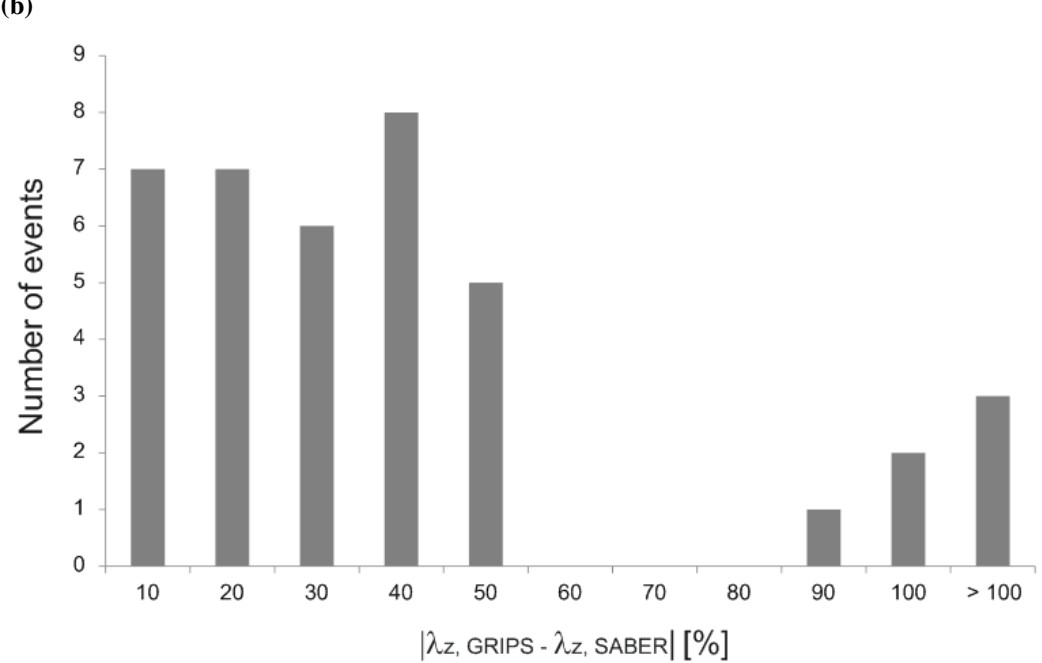

5   **Figure 4 The mean difference between the vertical wavelength derived by the scanning GRIPS $\lambda_{z,GRIPS}$ and the one identified in collocated TIMED-SABER vertical temperature profiles $\lambda_{z,SABER}$ is ca. 4.5 km (a) or 41% (b). In few cases, the difference is larger than 50%.**



**Table captions**

1      Brunt-Väisälä frequency for the different months at the OH-layer height above Oberpfaffenhofen

2      Period T and vertical wavelength $\lambda z$ derived from GRIPS measurements during different nights (DoY = day of year

5      when the measurement started), and date as well as time of the co-located SABER measurements.



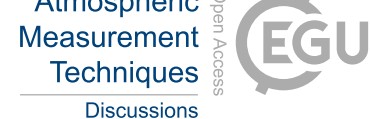

**Table 1**

| Month | N [1/s] |
|---|---|
| January | 0.0211 |
| February | 0.0211 |
| March | 0.0213 |
| April | 0.0221 |
| May | 0.0234 |
| June | 0.0244 |
| July | 0.0241 |
| August | 0.0230 |
| September | 0.0218 |
| October | 0.0212 |
| November | 0.0210 |
| December | 0.0212 |



**Table 2**

| No. | DoY GRIPS (evening) | DoY SABER | UTC SABER | T [min] | λz [km] |
|---|---|---|---|---|---|
| 1 | 212 | 212 | 23.62 | 74 | 11.0 |
| 2 | 212 | 213 | 1.32 | 74 | 10.2 |
| 3 | 214 | 214 | 22.37 | 436 | 14.6 |
| 4 | 214 | 214 | 22.39 | 436 | 14.6 |
| 5 | 214 | 214 | 24.08 | 436 | 12.3 |
| 6 | 214 | 214 | 24.10 | 436 | 12.3 |
| 7 | 214 | 215 | 1.78 | 436 | 8.8 |
| 8 | 215 | 215 | 22.61 | 348 | 18.8 |
| 9 | 215 | 215 | 22.63 | 348 | 18.8 |
| 10 | 215 | 215 | 24.31 | 348 | 15.7 |
| 11 | 215 | 215 | 24.32 | 348 | 15.7 |
| 12 | 215 | 216 | 2.01 | 348 | 12.2 |
| 13 | 215 | 216 | 2.03 | 348 | 12.2 |
| 14 | 215 | 215 | 22.61 | 107 | 15.0 |
| 15 | 215 | 215 | 22.63 | 107 | 15.0 |
| 16 | 215 | 215 | 24.31 | 107 | 9.6 |
| 17 | 215 | 215 | 24.32 | 107 | 9.6 |
| 18 | 215 | 216 | 2.01 | 107 | 6.6 |
| 19 | 215 | 216 | 2.03 | 107 | 6.6 |
| 20 | 216 | 216 | 22.86 | 168 | 13.4 |
| 21 | 216 | 217 | 0.56 | 168 | 15.1 |
| 22 | 221 | 221 | 22.33 | 393 | 12.6 |
| 23 | 221 | 221 | 24.04 | 393 | 1.3 |
| 24 | 221 | 221 | 22.33 | 148 | 18.2 |
| 25 | 221 | 221 | 24.04 | 148 | 9.2 |
| 26 | 225 | 225 | 19.88 | 274 | 15.5 |
| 27 | 225 | 225 | 21.58 | 274 | 14.1 |
| 28 | 225 | 225 | 23.27 | 274 | 18.0 |
| 29 | 225 | 225 | 23.28 | 274 | 18.0 |



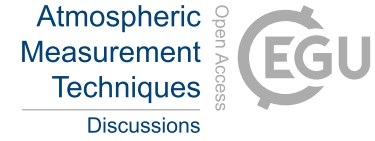

| 30 | 234 | 234 | 20.29 | 436 | 1.8 |
|----|-----|-----|-------|-----|-----|
| 31 | 234 | 234 | 21.99 | 436 | 5.0 |
| 32 | 239 | 239 | 19.77 | 536 | 1.2 |
| 33 | 239 | 239 | 21.46 | 536 | 6.1 |
| 34 | 254 | 254 | 23.13 | 626 | 5.8 |
| 35 | 254 | 254 | 23.14 | 626 | 5.8 |
| 36 | 259 | 259 | 22.59 | 324 | 38.0 |
| 37 | 259 | 259 | 22.60 | 324 | 38.0 |
| 38 | 259 | 259 | 22.59 | 200 | 8.6 |
| 39 | 259 | 259 | 22.60 | 200 | 8.6 |
| 40 | 263 | 263 | 21.82 | 608 | 14.3 |
| 41 | 263 | 263 | 21.83 | 608 | 14.3 |
| 42 | 275 | 275 | 19.53 | 616 | imaginary |
| 43 | 275 | 275 | 19.54 | 616 | imaginary |
| 44 | 275 | 275 | 19.53 | 158 | 45.9 |
| 45 | 275 | 275 | 19.54 | 158 | 45.9 |
| 46 | 284 | 285 | 3.51 | 557 | 6.7 |
| 47 | 284 | 285 | 3.51 | 257 | 28.6 |
| 48 | 292 | 293 | 1.99 | 550 | 12.3 |