# Peer review of "Derivation of horizontal and vertical wavelengths using a scanning OH(3-1) airglow spectrometer"

_Atmospheric Measurement Techniques, 2017_

## Referee Comment (RC1) · Anonymous Referee #1 · 3 Nov 2017

This manuscript describes the investigation of medium and long period gravity waves (GWs) observed at mesospheric altitude using a spectrometer instrument. Though this instrument only measures temperature variations within a limited field-of-view, it is possible to assess GW horizontal parameters by looking in 3 or 4 different directions. This technique has been previously published. The authors analysed 22 nights of data obtained from a mid-latitude site between July and November 2015. Using meteor wind radar data, they calculated the vertical wavelengths and compared their results with SABER observations. This paper is clear and well-written, nevertheless, I would suggest that the authors address the following comments: - The title should be changed to: "Derivation of mesospheric gravity waves horizontal and vertical wavelengths using..." - The error on the very long horizontal wavelengths must be really large. Wachter et al.,

2015 give Lx up to ∼1300 km and obtain already large uncertainties. I don't think values >1500 km make any sense. You should limit your study to the events with Lx<1500 km. - Many papers using airglow imagers to measure medium scale GWs are not mentioned in this paper: Takahashi et al., 2009; Paulino et al., 2011; 2012; Suzuki et al., 2013; Liu et al., 2015. Chen et al., 2013, and 2016 also investigated mesospheric large scale waves or inertial GWs using Fe lidar and radar data. The authors might not cite all these references (some of them only concern individual cases), but at least they should be aware of them. - What is the largest time difference between SABER measurements and GRIPS measurements? - Maybe you should have an extra figure to show the geometry of the observations. Something similar to Wachter et al., Figure 1, but for the configuration used in this study. - Table 2 should include the other parameters: Lh and c (and maybe also direction of propagation and wind speed in the GW direction).

Minor points: p. 1: l. 18: mesopause level or altitude l. 20: frequencies l. 21: remove "afterwards" l. 22: ...Oberpfaffenhofen, by a meteor radar. p. 2: l. 2: "is observed" or "is monitored" instead of "is addressed" l. 9: something wrong with this sentence l. 14: a few 100s km l. 15: of a few 10s km l. 31: constructed p. 3: l. 2: "operates" instead of "measures" p. 4: l. 27: "um" has to be changed with micron character (maybe it's just a problem of conversion to pdf format) p. 5: l. 24: 2-element l. 33: that applied p. 6: l. 3: components p. 7: l. 11: the maximum measurable period should be half the measurement time l. 21: 3600km is huge!!!! p. 8: l. 6: For medium and low-frequency waves, you always have N»sigma, so it works for your approximation, but maybe you don't need to talk about f since you don't use its relation with sigma or N. You should also explain why you can get rid of 1/4H2 l. 14: monthly basis p. 10: l. 2: change subtitle to 4.1 Horizontal parameters l. 20: the night p. 11: l. 6: the value for the scale height is surprising, usually at this altitude it's 6-7 km l. 8: investigated p. 12: l. 2: remove "," after only l. 3: ... and subsequently of the vertical... p. 13: l. 3: (1-10h, 100-1000s km)

The authors should use other expressions for "ca." (about, approximately, ~). It's a little bit repetitive!

Takahashi, H., M.J. Taylor, P.-D. Pautet, A.F. Medeiros,D. Gobbi, C.M. Wrasse, J. Fechine, M.A. Abdu, I.S. Batista, E. Paula, J.H.A. Sobral, D. Arruda, S.L. Vadas, F. Sao Sabbas, and D.C. Fritts, Simultaneous observation of ionospheric plasma bubbles and mesospheric gravity waves during the SpreadFEx Campaign, Ann. Geophys., 27,1477–1487, 2009

Paulino, I., H.Takahashi, A.F.Medeiros, C.M.Wrasse, R.A.Buriti, J.H.A.Sobral, and D.Gobbi, Mesospheric gravity waves and ionospheric plasma bubbles observed during the COPEX campaign, J. Atmos. Sol.-Terr. Phys., 73, 1575–1580, 2011

Paulino, I., H. Takahashi, S.L. Vadas, C.-M. Wrasse, J.H.A. Sobral, A.F. Medeiros, E. Buriti, and D. Gobbi, Forward ray-tracing for medium scale gravity waves observed during the COPEX campaign, J. Atmos. Sol.-Terr. Phys., 90-91, 117 – 123, 2012.

Suzuki, S., F.-J. Lubken, G. Baumgarten, N. Kaifler, R. Eixmann, B.P. Williams, and T. Nakamura, Vertical propagation of a mesoscale gravity wave from the lower to the upper atmosphere, J. Atmos. Sol.-Terr. Phys. , 97, 29–36, 2013

Liu X., C. Chen, W. Huang, J.A. Smith, X. Chu, T. Yuan, P.-D. Pautet, M.J. Taylor, J. Gong, and C.H. Cullens, A coordinated study of 1-h mesoscale gravity waves propagating from Logan to Boulder with CRRL Na Doppler lidars and temperature mapper, J. Geophys. Res. Atmos., 120, 10,006–10,021, doi: 10.1002/2015JD023604, 2015

Chen, C., X. Chu, A.J. McDonald, S.L. Vadas, Z. Yu, W. Fong, and X. Lu, Inertia-gravity waves in Antarctica: A case study using simultaneous lidar and radar measurements at McMurdo/Scott Base (77.8°S, 166.7°E), J. Geophys. Res., 118, 7, 2794–2808, DOI: 10.1002/jgrd.50318, 2013.

Chen, C., X. Chu, J. Zhao, B.R. Roberts, Z. Yu, W. Fong, X. Lu, and J.A. Smith, Lidar observations of persistent gravity waves with periods of 3–10?h in the Antarctic middle

and upper atmosphere at McMurdo (77.83°S, 166.67°E), J. Geophys. Res., 121, 2, 1483–1502, DOI: 10.1002/2015JA022127, 2016.
* * *

---

## Referee Comment (RC2) · Anonymous Referee #2 · 7 Nov 2017

This work is an analysis of gravity wave horizontal and vertical wavelength based on a spectroscopy measurement of OH vibrational temperature at 4 directions, 3 off-zenith and 1 at zenith. The horizontal wavelengths were derived based on phase differences among different directions. The vertical wavelength was derived based on gravity wave linear theory, and a nearby meteor radar wind measurements, together with assumed climatological Brunt-Väisälä frequency. The derived vertical wavelength was also compared with those derived from SABER temperature profiles. Error analysis of vertical wavelength was performed based on estimated errors of other parameters in the gravity wave dispersion relation.

[Figure]

The major additional work from a previous 3-direction measurement is to the derive vertical wavelength. This, in my opinion, does not provide any useful information. The derived vertical wavelengths have large errors, and have no consistency when compared with SABER. The measurement of a single airglow layer temperature can perhaps derive horizontal wavelength and period, when done very carefully, but it does not actually provide any information on the vertical wavelength. The inferred vertical wavelength is critically based on wind measurement elsewhere, and the dispersion relation. The major contribution from the OH measurement in addition to the wave period is the horizontal wavelength, which has already be published. Hence, I do not see value in publishing this work. If the authors like to improve on the current work, I'd recommend addressing on the improvement that a 4-direction measurement can make over the previous 3-direction measurement. In the following I list several major problems with vertical wavelength derivations, which I think cannot be mitigated because it's intrinsic to the limit of the measurements.

The mean difference of 4 km (Fig.4), over a mean vertical wavelength of 12.5 km (Fig.2) is quite large. The fact that the OH vertical wavelength has an error of 59%, ever larger than the mean difference from SABER (at 41%) means the comparison with SABER is meaningless. The difference cannot possibility be smaller than the error. It shows that these values are purely incidental.

page 9: The possibility that the intrinsic frequency is very close to f is not addressed. Since the uncertainties in u and v is 20 m/s, and the measured phase velocities are mostly between 20-40 m/s (10,5-6), some waves could have very small intrinsic phase speed and frequency. When this happens, all the errors in eqs.(4)-(8) will be huge, and the linear approximation in the error analysis does not apply anymore.

How will the emission height affect the derived horizontal wavelength, and thus vertical wavelength? Its effect is not considered in the error analysis. The height is known to vary, especially with tidal motion, by several km.
page 4, line 12: Since the 4 directions were measured at different times, how do this affect the derivation of horizontal wavelength? Is the 15 s or longer lag between different directions taken into account? How does this affect the errors?

page 5, line 6,11: 60-80 km temperature from SABER is used to derive vertical wavelength, but this region does not overlap with the airglow region. The reason given is that the temperature is sharply changing around 86 km in summer. This shows a problem with deriving vertical wavelength using a snap shot of temperature profile. Without temporal revolution, one cannot tell whether a sharp gradient is due to a wave or a more permanent feature. Therefore, the SABER derived vertical wavelength itself is not reliable either. Furthermore, if a large gradient does exist in the airglow region, the vertical wavelength derived below 80 km is not a reliable comparison, because waves propagating into a large N square region (a strong inversion layer below the mesopause) will change vertical wavelength.

page 2, line 25: Only one OH spectrometer, is in contrast with .., but implies that no other measurements are needed. This is not true, since meteor radar wind was used, and N2 is approximated.

page 4,line 10-11: not clear what the FoV size of the triangle means. They are several hundred km, much larger than the length of the triangle.

Table 2 only lists SABER vertical wavelength. Why not put the OH vertical wavelength as an additional column for direct comparison?

Why use monthly mean N square, not use the SABER temperature to derive N square?

page 2, line 15: 'maximum' is ambiguous. I suppose this means 'maximum number of waves'

---

## Referee Comment (RC3) · Anonymous Referee #3 · 9 Nov 2017

This paper derives spatial information on gravity waves near the menopause using airglow observations made at locations made at widely spaced positions in the horizontal. Using wind measurements made with a meteor-wind radar it is possible to convert the ground-based frequencies of the waves observed with airglow to intrinsic frequencies (frequencies measured in a coordinate system moving with the background wind) to infer important quantities such as GW vertical wavelength, that otherwise cannot be made with conventional single optical wavelength airglow observations. Results are compared with estimates based on TIMED-SABER data acquired on near-time overpasses. It is good to see comprehensive error estimates.

The paper is reasonably well written, although many sentences are convoluted. The paper would benefit from strong editing to improve readability and impact. Overall, the

results are relatively unique and merit publication in AMT, subject to improvements in language and clarification of some issues that are unclear.

When talking about wave periods or frequencies it should be made clear whether these are ground-based or intrinsic e.g. Abstracts, line 17 add "ground-based" before "periods".

2. P2, L2, "is studied using" is better grammatically than ""is addressed by"

3. P2, L9, The sentence starting "A sufficient number . . ." doe not make sense and needs rewriting.

4. L12, delete "under".

5. L16, delete "like they"

6. L29, "intrinsic frequency" not defined. Define here rather than later in the paper.

7. P4, section 21. The discussion of the FoV of the instrument is confusing. It is stated that the FoV triangle has an edge length of 90 km, but then it is stated that the FoV sizes are approximately 880 km. In what direction? If the field of view is 880 km x 880 km doesn't this average out all the detail in wave field? Schmidt et al (2013) claims that the GRIPS instrument has a FoV at 90 km equivalent to just a few km, which seems more reasonable. Please clarify.

8. L9, add "the" before "zenith".

9. P5, the tense of "data' is confusing - singular in L4, plural in L6. Plural is better.

10. P7, L19, "insert "the" before literature and move "not" from before "exceed" to after "do".

11. L22, delete superfluous "steps".

12. P8, L4, insert "density" before "scale height".

13. L6, the order of "medium and low-frequency waves" should be reversed to correspond to the order of the bracketed definitions. i.e. low- and medium frequency waves
($\sigma \sim$f ...)

14. Why not use the SABER temperature data to derive N? This will be more accurate
than using outdated monthly mean CIRA-86 temperature profiles.

15. P10, L18. The sentence starting :Depending on .." is confusing and needs re-
writing.

16. L29, "twice" rather than "two times"

17. Table 2, is T the ground-based or intrinsic period? Define.

---

## Author Comment (AC1) · 19 Dec 2017

We would like to thank the anonymous referee for his valuable comments. We tried to include them all. Please find our comments below.

Due to the comments of all three reviewers, I made the following general changes in the manuscript:

- The calculation of the vertical wavelengths from SABER data was limited to one wavelength for each profile in the range of the vertical wavelength derived from GRIPS +/- the error. As I re-calculated the approximation for the height range 70–90 km (instead of 60–80 km) for comparison reasons, I found out that the original approach

might deliver not the best results. The SABER profiles show two to three waves. If I restrict the adaption to one more or less specific oscillation, which might not be the dominant one, the harmonic analysis provides a kind of compromise between both waves. Therefore, I provided less restrictions to the harmonic analysis: it searched for two oscillations with a wavelengths between 2.5 km (minimal vertical wavelengths detectable in SABER measurements according to Trinh et al., 2015) and 20 km (height interval length) and I used the one which fits better to the GRIPS vertical wavelength. Applying this approach, the difference between the vertical wavelengths derived from both approaches halves.

- When adding additional information to former table 2, I found out, that I included one wave with a rather long wavelength (33 km) in the subsequent analysis. This is not consistent with the exclusion of waves with vertical wavelengths longer than 20 km. Therefore, I corrected it.
- I used the Brunt-Vaisala frequency calculated directly from the SABER profiles

These leads to different figures compared to the previous version. However, the main message of the paper does not change.

[Figure]

This manuscript describes the investigation of medium and long period gravity waves (GWs) observed at mesospheric altitude using a spectrometer instrument. Though this instrument only measures temperature variations within a limited field-of-view, it is possible to assess GW horizontal parameters by looking in 3 or 4 different directions. This technique has been previously published. The authors analysed 22 nights of data obtained from a mid-latitude site between July and November 2015. Using meteor wind radar data, they calculated the vertical wavelengths and compared their results with SABER observations. This paper is clear and well-written, nevertheless, I would suggest that the authors address the following comments:

- The title should be changed to: "Derivation of mesospheric gravity waves horizontal and vertical wavelengths using..." Done.
- The error on the very long horizontal wavelengths must be really large. Wachter et al., 2015 give Lx up to ~1300 km and obtain already large uncertainties. I don't think val- ues >1500 km make any sense. You should limit your study to the events with Lx<1500 km.

Wavelengths larger than 1500 km are only addressed in three cases (2x1801 km, 1x2054 km). Therefore, their effect on the mean values is not very large. The errors are approximately 420 km and 580 km (ca. 20–30%) in these cases.
I checked the analyses results; the mentioned cases do not produce outliers and the large error bars are taken into account for the calculation of the error bar of the vertical wavelength. Therefore, I included your suggestion as follows: in the text, I provided the (mean) results for waves with horizontal wavelengths of 1500 km at maximum, the (mean) values including all events are given in brackets, if they disagree. In the figures, the values referring to waves with horizontal wavelengths larger than 1500 km are marked in light grey.

- Many papers using airglow imagers to measure medium scale GWs are not mentioned in this paper: Takahashi et al., 2009; Paulino et al., 2011; 2012; Suzuki et al., 2013; Liu et al., 2015. Chen et al., 2013, and 2016 also investigated mesospheric large scale waves or inertial GWs using Fe lidar and radar data. The authors might

not cite all these references (some of them only concern individual cases), but at least they should be aware of them. Thank you for this hint. I included the ones which use a larger data basis (Paulino et al., 2011 and Che et al., 2016).

- What is the largest time difference between SABER measurements and GRIPS measurements? The GRIPS measurements are performed during night, therefore the use of SABER data is limited to nightly satellite overpasses. The exact times are given in table 1 (former table 2). The length of a GRIPS measurement depends on the length of the night and therefore on the day of the year and on the meteorological conditions. In our study, we only used nights with a very good signal to noise ratio (in order to identify the phase shift properly) with 7 h measurement time at minimum. The SABER measurements took place while GRIPS measured.

- Maybe you should have an extra figure to show the geometry of the observations. Something similar to Wachter et al., Figure 1, but for the configuration used in this study. I inserted such a figure as new figure 1 and changed the text in section 2.1 slightly (description of GRIPS).

- Table 2 should include the other parameters: Lh and c (and maybe also direction of propagation and wind speed in the GW direction).
I included two further parameters, horizontal wavelength derived by GRIPS and vertical wavelength derived by SABER. I provided exclusively these two in order to keep it clear. However, they can be used for the derivation of additional information like the phase velocity.

[Figure]

Minor points:

p. 1:

l. 18: mesopause level or altitude Done

l. 20: frequencies Done

l. 21: remove "afterwards" Done

l. 22: ...Oberpfaffenhofen, by a meteor radar. Done

p. 2:

l. 2: "is observed" or "is monitored" instead of "is addressed" I took "studied", it was the proposition of another reviewer whose corrections I read earlier.

l. 9: something wrong with this sentence Corrected

l. 14: a few 100s km Done

l. 15: of a few 10s km Done

l. 31: constructed Done

p. 3: l. 2: "operates" instead of "measures" Done

p. 4: l. 27: "um" has to be changed with micron character (maybe it's just a problem of conversion to pdf format) Done

p. 5:

l. 24: 2-element Done

l. 33: that applied I am not a native speaker but are you sure? Shouldn't it be "the same than applied" or "the same that was applied"?

p. 6: l. 3: components Done

p. 7:

l. 11: the maximum measurable period should be half the measurement time

If we used the FFT, I would agree with you. However, we are using the harmonic analysis. Here, the spectral resolution is not determined by the length of the data series. In principal, it would be sufficient, if half the oscillation was included in the time series. However, since we use the phase information, we need to be more careful here and use the full length of the data series. In addition, we apply several further criteria in order to check the consistency of the results based on this analysis step (see section 3.1).

Another point is that the harmonic analysis assumes a stationary signal. If the oscillation is much shorter than the time series and not stationary during the measurement time, the results (amplitude, phase and period) become uncertain, too.

So, the choice of these criteria is a compromise.

However, at least on average the measurement time is a little bit more than twice the period.

l. 21: 3600km is huge!!!! See comment above

p. 8: l. 6: For medium and low-frequency waves, you always have N»sigma, so it works for your approximation, but maybe you don't need to talk about f since you don't use its relation with sigma or N. You should also explain why you can get rid of 1/4H2

I inserted "The term $\frac{1}{4H^2}$ can be neglected since it is small compared to the squared vertical wave number".

l. 14: monthly basis Done (I corrected this mistake also in section 4.2, last paragraph before section 5)

p. 10:

[Figure]

l. 2: change subtitle to 4.1 Horizontal parameters Done

l. 20: the night Done

p. 11:

l. 6: the value for the scale height is surprising, usually at this altitude it's 6-7 km To

```
JANUARY   ZONAL MEAN TEMPERATURE (K)

Scale Pressure Geom.
Height  (mb)  Height   0    5N   10
              (km)

13.00 2.29E-3  89.2  190.1 190.2 190
12.75 2.94E-3  87.7  192.7 192.9 193
12.50 3.78E-3  86.3  195.7 196.0 196
```

To be honest, I was not able to read the CIRA properly. I am sorry. Even if the parameter listed in the first column (see figure above) is called scale height, it is the log-pressure scale height ($-\ln(p/p_0)$). Therefore, I deleted this part of the manuscript.

l. 8: investigated Done

p. 12:

l. 2: remove "," after only Done

l. 3: ... and subsequently of the vertical... Done

p. 13: l. 3: (1-10h, 100-1000s km) Done

The authors should use other expressions for "ca." (about, approximately, ~). It's a little bit repetitive! Indeed, we use it 29 times. I tried to mix it a little bit.

[Figure]

Takahashi, H., M.J. Taylor, P.-D. Pautet, A.F. Medeiros,D. Gobbi, C.M. Wrasse, J. Fechine, M.A. Abdu, I.S. Batista, E. Paula, J.H.A. Sobral, D. Arruda, S.L. Vadas, F. Sao Sabbas, and D.C. Fritts, Simultaneous observation of ionospheric plasma bubbles and mesospheric gravity waves during the SpreadFEx Campaign, Ann. Geophys., 27,1477–1487, 2009

Paulino, I., H.Takahashi, A.F.Medeiros, C.M.Wrasse, R.A.Buriti, J.H.A.Sobral, and D.Gobbi, Mesospheric gravity waves and ionospheric plasma bubbles observed during the COPEX campaign, J. Atmos. Sol.-Terr. Phys., 73, 1575–1580, 2011

Paulino, I., H. Takahashi, S.L. Vadas, C.-M. Wrasse, J.H.A. Sobral, A.F. Medeiros, E. Buriti, and D. Gobbi, Forward ray-tracing for medium scale gravity waves observed during the COPEX campaign, J. Atmos. Sol.-Terr. Phys., 90-91, 117 – 123, 2012.

Suzuki, S., F.-J. Lubken, G. Baumgarten, N. Kaifler, R. Eixmann, B.P. Williams, and T. Nakamura, Vertical propagation of a mesoscale gravity wave from the lower to the upper atmosphere, J. Atmos. Sol.-Terr. Phys. , 97, 29–36, 2013

Liu X., C. Chen, W. Huang, J.A. Smith, X. Chu, T. Yuan, P.-D. Pautet, M.J. Taylor, J. Gong, and C.H. Cullens, A coordinated study of 1-h mesoscale gravity waves propagating from Logan to Boulder with CRRL Na Doppler lidars and temperature mapper, J. Geophys. Res. Atmos., 120, 10,006–10,021, doi: 10.1002/2015JD023604, 2015

Chen, C., X. Chu, A.J. McDonald, S.L. Vadas, Z. Yu, W. Fong, and X. Lu, Inertia-gravity waves in Antarctica: A case study using simultaneous lidar and radar measurements at McMurdo/Scott Base (77.8°S, 166.7°E), J. Geophys. Res., 118, 7, 2794–2808, DOI: 10.1002/jgrd.50318, 2013.

Chen, C., X. Chu, J. Zhao, B.R. Roberts, Z. Yu, W. Fong, X. Lu, and J.A. Smith, Lidar observations of persistent gravity waves with periods of 3–10?h in the Antarctic middle

and upper atmosphere at McMurdo (77.83◦S, 166.67◦E), J. Geophys. Res., 121, 2, 1483–1502, DOI: 10.1002/2015JA022127, 2016.

---

## Author Comment (AC2) · 19 Dec 2017

We would like to thank Alan Liu for his valuable comments. We answered all of them and changed the manuscript where we think it is necessary. Please find our list of answers and comments below.

Due to the comments of all three reviewers, I made the following general changes in the manuscript:

- The calculation of the vertical wavelengths from SABER data was limited to

one wavelength for each profile in the range of the vertical wavelength derived from GRIPS +/- the error. As I re-calculated the approximation for the height range 70–90 km (instead of 60–80 km) for comparison reasons, I found out that the original approach might deliver not the best results. The SABER profiles show two to three waves. If I restrict the adaption to one more or less specific oscillation, which might not be the dominant one, the harmonic analysis provides a kind of compromise between both waves. Therefore, I provided less restrictions to the harmonic analysis: it searched for two oscillations with a wavelengths between 2.5 km (minimal vertical wavelengths detectable in SABER measurements according to Trinh et al., 2015) and 20 km (height interval length) and I used the one which fits better to the GRIPS vertical wavelength. Applying this approach, the difference between the vertical wavelengths derived from both approaches halves.

- When adding additional information to former table 2, I found out, that I included one wave with a rather long wavelength (33 km) in the subsequent analysis. This is not consistent with the exclusion of waves with vertical wavelengths longer than 20 km. Therefore, I corrected it.
- I used the Brunt-Vaisala frequency calculated directly from the SABER profiles

These leads to different figures compared to the previous version. However, the main message of the paper does not change.

This work is an analysis of gravity wave horizontal and vertical wavelength based on a spectroscopy measurement of OH vibrational temperature at 4 directions, 3 off-zenith and 1 at zenith. The horizontal wavelengths were derived based on phase differences among different directions. The vertical wavelength was derived based on gravity wave linear theory, and a nearby meteor radar wind measurements,

together with assumed climatological Brunt-Väisälä frequency. The derived vertical wavelength was also compared with those derived from SABER temperature profiles. Error analysis of vertical wavelength was performed based on estimated errors of other parameters in the gravity wave dispersion relation.

The major additional work from a previous 3-direction measurement is to the derive vertical wavelength. This, in my opinion, does not provide any useful information. The derived vertical wavelengths have large errors, and have no consistency when compared with SABER. The measurement of a single airglow layer temperature can perhaps derive horizontal wavelength and period, when done very carefully, but it does not actually provide any information on the vertical wavelength. The inferred vertical wavelength is critically based on wind measurement elsewhere, and the dispersion relation. The major contribution from the OH measurement in addition to the wave period is the horizontal wavelength, which has already be published. Hence, I do not see value in publishing this work. If the authors like to improve on the current work, I'd recommend addressing on the improvement that a 4-direction measurement can make over the previous 3-direction measurement. In the following I list several major problems with vertical wavelength derivations, which I think cannot be mitigated because it's intrinsic to the limit of the measurements.

The mean difference of 4 km (Fig.4), over a mean vertical wavelength of 12.5 km (Fig.2) is quite large. The fact that the OH vertical wavelength has an error of 59%, ever larger than the mean difference from SABER (at 41%) means the comparison with SABER is meaningless. The difference cannot possibility be smaller than the error. It shows that these values are purely incidental. The estimation of the error is

based on error propagation calculation. Here, the uncertainties of the different parameters contribute. Even if done very carefully, these uncertainties are still estimates. The comparison of the vertical wavelengths based on the spectrometer radar combination to the vertical wavelengths derived from SABER shows that the provided error is in the same order of magnitude but might be too conservative. We changed "A mean error of 59% as calculated based on error propagation can be therefore regarded as a conservative estimate." to "The vertical wavelengths agree within the error bars in all but four cases; here, the vertical wavelengths derived from SABER are slightly smaller by 0.2–0.8 km."

page 9: The possibility that the intrinsic frequency is very close to f is not addressed.

It does not need to be addressed since it is not the case. It can happen that the intrinsic frequency and f are similar, however, these values are excluded from further analysis since the vertical wavelengths are rather small (as described in section 4.2).

Since the uncertainties in u and v is 20 m/s, and the measured phase velocities are mostly between 20-40 m/s (10,5-6), some waves could have very small intrinsic phase speed and frequency. When this happens, all the errors in eqs.(4)-(8) will be huge, and the linear approximation in the error analysis does not apply anymore.

How will the emission height affect the derived horizontal wavelength, and thus vertical wavelength? Its effect is not considered in the error analysis. The height is known to vary, especially with tidal motion, by several km.

Such effects are of interest, if they influence the different FoV to a different extent.

[Figure]

Our results rely on spatial and temporal averages:

- The FoV covers 880 km² (560 km²), all values, we derive are averaged over this area.
- We analyse time series of 7 h and longer.

Furthermore, the four FoV are rather near to each other (see new figure 1 for better understanding). This reduces the possible effect of large scale motions like tides on our analysis results tremendously.

Third, due to the redundancy of the system (we get four values for the horizontal wavelength), we dismiss results which do not agree sufficiently (as mentioned in section 3.1). This might be the case when not all but only one or two FoV are influenced by a higher or lower airglow altitude.

Therefore, we judge the effect the reviewer mentions as an effect of second-order and neglect it.

page 4, line 12: Since the 4 directions were measured at different times, how do this affect the derivation of horizontal wavelength? Is the 15 s or longer lag between different directions taken into account? How does this affect the errors? As mentioned on page 2 "The instrument acquires spectra with a temporal resolution of 15 s. … In order to improve the signal-to-noise ratio for the intended analysis five minute mean values are calculated for each FoV." Furthermore, we write on page 7 "Further analysis steps are restricted to results which are characterized by a period longer (shorter) than 60 min (the measurement time)". Therefore, we assume that the time lag of 15 s is negligible.

[Figure]

page 5, line 6,11: 60-80 km temperature from SABER is used to derive vertical wavelength, but this region does not overlap with the airglow region. The reason given is that the temperature is sharply changing around 86 km in summer. This shows a problem with deriving vertical wavelength using a snap shot of temperature profile. Without temporal revolution, one cannot tell whether a sharp gradient is due to a wave or a more permanent feature. Therefore, the SABER derived vertical wavelength itself is not reliable either.

In order to analysis a temporal-spatial phenomenon such as a wave, I should ideally use data which are provided with sufficient temporal and spatial (3D) resolution. However, such data are not available. Based on this argumentation, I can never use data of orbiting satellites or of radiosondes, for example. They are always snapshots and the repetition rate of these instruments is far too low compared to the lifetime of gravity waves. Also lidar data could only be used in parts since they do not provide horizontally-resolved data. In principal, I can expand this argumentation to every instrument.

Since we are aware of the fact that the mesopause might cause a problem in detrending, we used the height range slightly below.

In addition, I repeated the analysis based on the height range of 70–90 km for our data covering September to December (=months with higher mesopause). The results look pretty similar.

Furthermore, if a large gradient does exist in the airglow region, the vertical wavelength derived below 80 km is not a reliable comparison, because waves propagating into a large N square region (a strong inversion layer below the

[Figure]

mesopause) will change vertical wavelength.

That's true, vertical wavelengths are always influenced by the wind field and the Brunt-Vaisala frequency which changes more or less with height. This effect weakens (averages out) when using temporal or spatial averages (e.g., due to the measurement technique or the analysis method). When I analysis the height range 60–80 km or 70–90 km and the wave changes its wavelengths there since the background atmosphere varies, I will get an averaged value, for example.

Therefore, we would never expect an ideal agreement between both data sets. We use the SABER data just as a kind of consistency check. The question we would like to answer is: do we find wavelengths in SABER data which agree with the wavelengths derived from the GRIPS-radar combination within the error bars. If we hadn't used the SABER data, we would probably have to answer the question from another reviewer why we did not look into other data sets to check our results. That is why we present the SABER analysis here, even if we know that this combination is not ideal.

page 2, line 25: Only one OH spectrometer, is in contrast with .., but implies that no other measurements are needed. This is not true, since meteor radar wind was used, and N2 is approximated. Changed to "Here, for the first time, we present an approach to derive zonal, meridional and vertical wavelengths as well as wave periods based on only one OH* spectrometer addressing one vibrational-rotational transition and on additional information about the horizontal wind and the Brunt-Väisälä frequency.

page 4,line 10-11: not clear what the FoV size of the triangle means. They are

[Figure]

several hundred km, much larger than the length of the triangle. Changed the formulation to "The edge length of the FoV triangle amounts to 90 km. Due to the finite aperture of the GRIPS 14, each FoV covers approximately 880 km² excluding the one in the zenith direction. The latter is smaller with ca. 560 km² (see fig. 1)." and added a sketch to make it clearer.

Table 2 only lists SABER vertical wavelength. Why not put the OH vertical wavelength as an additional column for direct comparison? Done

Why use monthly mean N square, not use the SABER temperature to derive N square? Done. I also inserted in section 3.2 why we still use a relative uncertainty of 10%.

page 2, line 15: 'maximum' is ambiguous. I suppose this means 'maximum number of waves' Yes and corrected

---

## Author Comment (AC3) · 19 Dec 2017

We would like to thank the anonymous referee for his valuable comments. We included all of them. Please find our comments below.

Due to the comments of all three reviewers, I made the following general changes in the manuscript:

- The calculation of the vertical wavelengths from SABER data was limited to one wavelength for each profile in the range of the vertical wavelength derived from GRIPS +/- the error. As I re-calculated the approximation for the height range 70–90 km (instead of 60–80 km) for comparison reasons, I found out that the original approach might deliver not the best results. The SABER profiles show two to three waves. If I restrict the adaption to one more or less specific oscillation, which might not be the dominant one, the harmonic analysis provides a kind of compromise between both waves. Therefore, I provided less restrictions to the harmonic analysis: it searched for two oscillations with a wavelengths between 2.5 km (minimal vertical wavelengths detectable in SABER measurements according to Trinh et al., 2015) and 20

km (height interval length) and I used the one which fits better to the GRIPS vertical wavelength. Applying this approach, the difference between the vertical wavelengths derived from both approaches halves.

- When adding additional information to former table 2, I found out, that I included one wave with a rather long wavelength (33 km) in the subsequent analysis. This is not consistent with the exclusion of waves with vertical wavelengths longer than 20 km. Therefore, I corrected it.
- I used the Brunt-Vaisala frequency calculated directly from the SABER profiles

These leads to different figures compared to the previous version. However, the main message of the paper does not change.

This paper derives spatial information on gravity waves near the menopause using airglow observations made at locations made at widely spaced positions in the horizontal. Using wind measurements made with a meteor-wind radar it is possible to convert the ground-based frequencies of the waves observed with airglow to intrinsic frequencies (frequencies measured in a coordinate system moving with the background wind) to infer important quantities such as GW vertical wavelength, that otherwise cannot be made with conventional single optical wavelength airglow observations. Results are compared with estimates based on TIMED-SABER data acquired on near-time over-passes. It is good to see comprehensive error estimates.

The paper is reasonably well written, although many sentences are convoluted. The paper would benefit from strong editing to improve readability and impact. Overall, the results are relatively unique and merit publication in AMT, subject to improvements in language and clarification of some issues that are unclear.

When talking about wave periods or frequencies it should be made clear whether these are ground-based or intrinsic e.g. Abstracts, line 17 add "ground-based" before "periods".Done, I also added this information in section 3.1

2. P2, L2, "is studied using" is better grammatically than ""is addressed by" Done

3. P2, L9, The sentence starting "A sufficient number . . ." doe not make sense and needs rewriting. Done

4. L12, delete "under". Done

5. L16, delete "like they" Done but inserted "and" before "cannot"

6. L29, "intrinsic frequency" not defined. Define here rather than later in the paper. Done

7. P4, section 21. The discussion of the FoV of the instrument is confusing. It is stated that the FoV triangle has an edge length of 90 km, but then it is stated that the FoV sizes are approximately 880 km. In what direction? If the field of view is 880 km x 880 km doesn't this average out all the detail in wave field? Schmidt et al (2013) claims that the GRIPS instrument has a FoV at 90 km equivalent to just a few km, which seems more reasonable. Please clarify.

I changed the formulation to "The edge length of the FoV triangle amounts to 90 km. Due to the finite aperture of the GRIPS 14, each FoV covers approximately 880 km² excluding the one in the zenith direction. The latter is smaller with ca. 560 km² (see fig. 1)." and added a sketch to make it clearer.

8. L9, add "the" before "zenith". Done

9. P5, the tense of "data' is confusing - singular in L4, plural in L6. Plural is better. Done

10. P7, L19, "insert "the" before literature and move "not" from before "exceed" to after "do". Done

11. L22, delete superfluous "steps". Done

12. P8, L4, insert "density" before "scale height". Done

13. L6, the order of "medium and low-frequency waves" should be reversed to correspond to the order of the bracketed definitions. i.e. low- and medium frequency waves ($\sigma \sim f \dots$) Done

14. Why not use the SABER temperature data to derive N? This will be more accurate than using outdated monthly mean CIRA-86 temperature profiles. Done. I also inserted in section 3.2 why we still use a relative uncertainty of 10%.

15. P10, L18. The sentence starting :Depending on .." is confusing and needs rewriting. Done

16. L29, "twice" rather than "two times" Done

17. Table 2, is T the ground-based or intrinsic period? Define. It's ground-based and I added this information

---

## Author Response (AR1)

[revised manuscript text omitted]

| | | | | | | | |
|---|---|---|---|---|---|---|---|
| 25 | 221 | 221 | 24.04 | 148 | 10.8 | 8.9 | 525 |
| 26 | 225 | 225 | 19.88 | 274 | 15.1 | 8.7 | 857 |
| 27 | 225 | 225 | 21.58 | 274 | 14.1 | 9.8 | 857 |
| 28 | 225 | 225 | 23.27 | 274 | 17.4 | 10.9 | 857 |
| 29 | 225 | 225 | 23.28 | 274 | 18.3 | 12.4 | 857 |
| 30 | 234 | 234 | 20.29 | 436 | 1.9 | --- | 801 |
| 31 | 234 | 234 | 21.99 | 436 | 5.0 | 5.5 | 801 |
| 32 | 239 | 239 | 19.77 | 536 | 1.3 | --- | 81 |
| 33 | 239 | 239 | 21.46 | 536 | 6.4 | 9.4 | 81 |
| 34 | 254 | 254 | 23.13 | 626 | 5.8 | 5.3 | 657 |
| 35 | 254 | 254 | 23.14 | 626 | 5.5 | 3.6 | 657 |
| 36 | 259 | 259 | 22.59 | 324 | 36.6 | --- | 1691 |
| 37 | 259 | 259 | 22.60 | 324 | 38.8 | --- | 1691 |
| 38 | 259 | 259 | 22.59 | 200 | 8.3 | 8.5 | 421 |
| 39 | 259 | 259 | 22.60 | 200 | 8.8 | 11.7 | 421 |
| 40 | 263 | 263 | 21.82 | 608 | 15.9 | 13.8 | 1801 |
| 41 | 263 | 263 | 21.83 | 608 | 14.0 | 17.4 | 1801 |
| 42 | 275 | 275 | 19.53 | 616 | imaginary | --- | 3157 |
| 43 | 275 | 275 | 19.54 | 616 | imaginary | --- | 3157 |
| 44 | 275 | 275 | 19.53 | 158 | 44.6 | --- | 634 |
| 45 | 275 | 275 | 19.54 | 158 | 44.4 | --- | 634 |
| 46 | 284 | 285 | 3.51 | 557 | 7.8 | 5.4 | 2054 |
| 47 | 284 | 285 | 3.51 | 257 | 33.4 | --- | 436 |
| 48 | 292 | 293 | 1.99 | 550 | 12.4 | 14.1 | 891 |

---

## Author Response (AR2)

**Report #1**

Submitted on 11 Jan 2018
Anonymous Referee #3

**Anonymous during peer-review: Yes** No
**Anonymous in acknowledgements of published article: Yes** No

**Recommendation to the editor**

**1) Scientific significance**
Does the manuscript represent a substantial contribution to scientific progress within the scope of this journal (substantial new concepts, ideas, methods, or data)?

Excellent **Good** Fair Poor

**2) Scientific quality**
Are the scientific approaches and applied methods valid? Are the results discussed in an appropriate and balanced way (consideration of related work, including appropriate references)?

Excellent Good **Fair** Poor

**3) Presentation quality**
Are the scientific results and conclusions presented in a clear, concise, and well structured way (number and quality of figures/tables, appropriate use of English language)?

Excellent Good **Fair** Poor

For final publication, the manuscript should be

**accepted as is**

**accepted subject to technical corrections**

accepted subject to **minor revisions**

reconsidered after **major revisions**

> I would like to review the revised paper.

> I am **not** willing to review the revised paper.

**rejected**

**Suggestions for revision or reasons for rejection (will be published if the paper is accepted for final publication)**

We thank the anonymous reviewer for his comments.

The paper is basically acceptable but needs careful proofreading to eliminate unnecessary mistakes. For example:
P4 11, 880 km^2, not 880 km. Also 560 km^2. See also, caption to Figure 1. That's confusing—two of three reviewers note that the square is not there. However, in my original version of the manuscript I submitted, it appears.
P10, 26. Table 1, not Table 2. Thanks, changed

[Figure]

**Report #2**

Submitted on 15 Jan 2018
Anonymous Referee #1

**Anonymous during peer-review: Yes** No
**Anonymous in acknowledgements of published article: Yes** No

**Recommendation to the editor**

**1) Scientific significance**
Does the manuscript represent a substantial contribution to scientific progress within the scope of this journal (substantial new concepts, ideas, methods, or data)?

Excellent **Good** Fair Poor

**2) Scientific quality**
Are the scientific approaches and applied methods valid? Are the results discussed in an appropriate and balanced way (consideration of related work, including appropriate references)?

**Excellent** Good Fair Poor

**3) Presentation quality**
Are the scientific results and conclusions presented in a clear, concise, and well structured way (number and quality of figures/tables, appropriate use of English language)?

**Excellent** Good Fair Poor

For final publication, the manuscript should be

**accepted as is**

**accepted subject to technical corrections**

accepted subject to **minor revisions**

reconsidered after **major revisions**

      I would like to review the revised paper.

      I am **not** willing to review the revised paper.

**rejected**

**Suggestions for revision or reasons for rejection (will be published if the paper is accepted for final publication)**

The paper has been clearly improved. Thank you for clarifying some points and accommodating for my suggestions. I still have a few minor edits:

We thank the reviewer for his comments.

p.2 l.18: introduced Thanks, changed
l.20: 100s m. I changed it to "in the order of some kilometres to a few 100s m" since Hannawald et al. (2016) and Sedlak et al (2016) address these two ranges. These orders of magnitude are still much smaller than the one addressed by Wachter et al. (2015), which is mentioned in the sentences before and to which Hannawald et al. (2016) and Sedlak et al (2016) are compared.
p.11 l.6: remove one ( Thanks, changed
p.12 l.1: data base I learned from another reviewer some time ago that it should be basis and not base (since base is something like a fundament, so something concrete, basis is something abstract). But who is right? Anyway, I am open for any improvements of my English.
p.13 p.13 l.7: 19-20 km Thanks, changed

Figure 1: …in 4 different directions… are larger (880 km2)… Thanks, changed

[Figure]

**Report #3**

Submitted on 15 Jan 2018
Referee #2: Alan Liu, liuz2@erau.edu

**Anonymous during peer-review:** Yes  **No Anonymous in acknowledgements of published article: Yes** No

**Recommendation to the editor**

**1) Scientific significance**
Does the manuscript represent a substantial contribution to scientific progress within the scope of this journal (substantial new concepts, ideas, methods, or data)?

Excellent **Good** Fair Poor

**2) Scientific quality**
Are the scientific approaches and applied methods valid? Are the results discussed in an appropriate and balanced way (consideration of related work, including appropriate references)?

Excellent Good **Fair** Poor

**3) Presentation quality**
Are the scientific results and conclusions presented in a clear, concise, and well structured way (number and quality of figures/tables, appropriate use of English language)?

Excellent **Good** Fair Poor

For final publication, the manuscript should be

**accepted as is**

accepted subject to **technical corrections** accepted subject to **minor revisions reconsidered after major revisions**

> **I would like to review the revised paper.**
> I am **not** willing to review the revised paper.

**rejected**

**Suggestions for revision or reasons for rejection (will be published if the paper is accepted for final publication)**
Review

This revised manuscript is improved from the first version. However, I still have some major concerns, with both how the results are presented, and how the analysis was done. In the following, I provided comments following the sequence of the manuscript, with major ones marked by '*'. I also commented to the authors' replies to my first review at the end.

We appreciate the valuable comments of Alan Liu. In the following, we answered all of them and changed the manuscript accordingly where needed.

* P2, L5-10: I understand that the authors try to state the different strengths of different instruments, but they are not described accurately. There needs to be a distinction between what parameters an instrument can directly measure or sensitive to, and what parameters can be derived or 'investigated' from the measurements. For example, lidars can measure vertical profiles so can directly measure vertical variations and provide vertical wavelength. But they can also be used to derive horizontal wavelength (e.g. as in Hu et al. 2002, Lu et al. 2009, Chen et al. 2013) for inertia-gravity waves. Airglow images can directly measure horizontal wavelength but can also be used to derive vertical wavelength when background horizontal wind. The main message here is that with enough known parameters, any instrument can be used to derive other missing gravity wave parameters. The distinctions among different instruments should be what they can directly measure, not what they can be used to derive. I re-formulated this passage.

* P2, L24-26: The real 'first time' here is only about deriving vertical wavelength. Mixing them with 'zonal, meridional' wavelengths is a bit misleading. As referenced by the authors, Wachter et al. (2015) have already shown the technique of deriving horizontal wavelength. Furthermore, as I pointed in the first review, the ability to derive vertical wavelength is partially contributable to the

spatial information resolved by the scanning spectrometer (not new) and partially contributable to   the horizontal wind from a nearby meteor radar (not this instrument). I suggest changing the title to 'Derivation of gravity wave intrinsic parameters and vertical wavelength using a single scanning …' Deriving intrinsic parameters (which the vertical wavelength depends on) using a single   spectrometer is really the main contribution, not the horizontal wavelength.

I agree and changed the title. I also re-formulated this passage slightly in order to emphasize the reviewer's point.

P2, L5, 'sensitive for' -> 'sensitive to' Thanks
P2, L12, 'exclusively' -> 'only'. 'Exclusively' means no other instrument can do it. Thanks, but I changed the paragraph and exclusively is not used any more
P2, L30, Wave vector is always related to intrinsic wave frequency, not just for low- and medium-frequency waves. Changed

P4, L11-12: missing the squares after 'km' You mean after 880 and 560? But there are squares in my version, also in the pdf, I just checked it.
P4, L14-15: provide an estimated uncertainty of the 5 min averaged data Done

P5, L25-27: The sentence 'The radar delivers …' describes only a very small part of the whole procedure of meteor radar wind retrieval. The main procedure is a fitting of all radial winds within a time-altitude bin in a least-square sense, assuming homogeneity within the bin. Since 'fit' and  'fitting technique' are mentioned in subsequent sentences, what the fit means need to be described  first. We changed this paragraph in order to make the procedure clearer and corrected an inaccuracy: the height gates refer to 4 km and not to 3 km. We apologize for that.

P6, L6: 'dynamical' -> 'dynamically'        Thanks

* P8, L17-18: Even if the uncertainty of ground frequency is negligible, the intrinsic frequency may  be not, because the latter depends on the wind in the direction of wave propagation. This Doppler  wind depends on both the magnitude and direction of the background wind as well as the direction  of the wave. In the analysis, the uncertainty in k and l are considered, but its indirect effect on the  wave direction thus the component of the background wind in the direction of propagation is not.  Please address this source of uncertainty.
I agree that the error of the intrinsic frequency may not be negligible since k, l, u, and v are probably not exact. However, when calculating the error of $\lambda_z$, we do this by taking into account the error of k, l, u, and v and the contribution of the respective variable to formula (2). This also includes the influence of these variables on the intrinsic frequency. Therefore, we take into account the error of the intrinsic frequency.

P9, L2: It's not clear what 'too conservative' means. Does it mean the 10% is an underestimate or overestimate? Please clarify. We mean that the uncertainty of 10% might be too high, so overestimated. In Wüst et al. (2017b), we calculated a "climatology" of the Brunt-Vaisala frequency over the Alpine region based on 14 years of TIMED-SABER measurements and adapted a combination of three sinusoidals to the data. This mean curve allows to calculate the Brunt-Vaisala value for every day. Ca. 98% of the Brunt-Vaisala values (for each day of the year) are located in a ±10%-interval around this mean curve. Since this interval comprises nearly all values and the variability is reduced in autumn and winter (a time period which we also address here), 10% might be too high.

P10,
L5-6: Please clarify if this phase velocity is relative to the ground or intrinsic. Also, clarify in Figure  2b and its caption. It's the velocity relative to the ground → clarified
L21-22: I don't understand why the authors say 'no vertical … profile is suitable' first and then 'more than one is available'. Please clarify. The sentences were changed into "Depending on the orbit of TIMED, it can happen that no vertical temperature profile is suitable one day. However, the other day also more than one profile can be available."
L23: 'In one case' -> shouldn't it be 'In some cases'? There seem to be 31-19=12 cases that are without wind. The given numbers are correct, it is more a formulation problem. As said above, SABER data are not always available. In these cases, we didn't calculate the vertical wavelengths. I changed the sentence referring to the SABER data and hope that the main message is now clear. However, I am open for improvements.
L24: 'referring to' -> 'on' I can change it but I have a question before: I am only used to the article "to" in combination with "refer". I just looked it up once again in the two most common English-German web-dictionaries and in the Oxford dictionary, everywhere "refer" is used together with "to". When do I use it with "on"?

P7, L9-12: I think it's important to point out that this method looks for a single oscillation  throughout an entire night (according to Wachter et al. 2015). If that's not the case, then the  duration of each wave event should be given. Readers normally assume that a wave event could  occur during part of a night but not the entire night, as I did at first.
We added the information that the oscillation needs to present throughout the whole night.

P10, L25-26: If I understand correctly, the authors used one detected wave from the airglow and applied multiple nearby SABER profiles and corresponding winds to obtain multiple vertical wavelengths. That's correct. This needs to be clarified in the text. Done (page 10, line 24–28). It's also not clear whether all SABER data selected are within that time period. The spatial range is described in P5L3 but not the temporal range. Information about the temporal range is now provided on page 5, lines 4–7.

P11, L3: 'table 2' should 'Table 1'. It appears the authors have deleted Table 2. That's correct, thank you

P11, L4: evanescent waves are not necessarily damped. I deleted damped.

P11, L21: Are there always 'Two wavelengths' in a SABER profile? Table 1 shows some SABER vertical wavelengths are over 15 km, which means there can only be one wavelength in the 60-80 km range. The algorithms was programmed that way that it always searched for two wavelengths and in all cases two wavelengths were identified. The given wavelength is the one which agrees best with the wavelengths derived from the GRIPS-radar combination. The procedure is explained on page 11, second half of the page. We added the information that only the value which agrees best is mentioned in table 1 (p.11, line 34).

* P11, L23-26: As the authors stated, SABER is sensitive only to oscillations with long horizontal wavelength. In Table 1, there are two cases (DOY 212 and 215) where the horizontal wavelength is only 200 km. It's true that SABER may be seeing at a horizontal angle not perpendicular to the wavefront, therefore the cancellation effect is not severe. This is however only a speculation but it can be easily demonstrated by additional analysis. Since both the wave direction and the direction of SABER sounding profile are known, their relative angles can be calculated to examine whether this speculation is correct. This needs to be done to verify if that's indeed the case.

I calculated the difference in longitude and latitude of two SABER profiles (at 86 km height) which were measured successively at DoY 212 and 215. So, one gets information about the flight direction of TIMED. The SABER instrument looks perpendicular to the flight direction. It can deliver information in the ascending and descending branch.

As long as SABER is in the same yaw cycle, its field of view can be oriented only in two ways at each latitude. With respect to the North, the absolute values are identical.

For our two examples, SABER is flying at an azimuth angle of -67° (at ca. 45°N) and 74° (at ca. 48°N). That means the fields of view are oriented mainly north-southward (23° and -16°).

On DoY 212, the wave fronts are moving at an azimuth angle of 77° +/- 10° (with respect to the North, clockwise). That means the wave fronts are mainly oriented northward (187° +/- 10°).

On DoY 215, the wave fronts are moving at an azimuth angle of 299° +/- 14°. That means the wave fronts are also mainly oriented northward (389° +/- 14° = 29° +/- 14°).

So, in both cases SABER is not measuring perpendicular to the wave fronts. In the worst case, the angle is ca. 45°.

P24: Table 1 caption should state that the period is 'ground-based.' as one reviewer asked. Adding a small subscript in the variable in the table header is not obvious to the readers. Changed

P25: Table 1 should list the intrinsic wave periods. In the manuscript, as it is, there is no information at all about the background wind and its effect. For this reason, at least the intrinsic period (and/or wind in the direction of the wave, intrinsic frequency) is necessary. Another column needed is the percentage uncertainties of airglow vertical wavelength as shown in Figure 3b. The figure only shows the distributions but one cannot tell which uncertainty is for which wavelength. Done.

Figure 1: Adding Figure 1 helps a lot. In the caption, the square after km is also missing. In my version, it is there. However, also one other review mentions this issue. Since the area of the zenith square is given, it seems natural to also give the area of the off-zenith areas in the caption. Done.

**Comments to the authors' replies: Regarding the tidal effect:**

The effects of tide are not about affecting different FOV differently, but about changing the assumed 86 km altitude of OH airglow. This has two effects on the derive vertical wavelength. One is changing the assumed distance between different FOVs, therefore, the derived horizontal wavelength. This error depends on the horizontal wavelength (the shorter the larger the error).

If the altitude of the FoV is changed by the same constant and not by an individually varying one (as discussed in review 1), this should not affect our results.

We measure the temperatures $T_1(x_1, y_1, z_1)$ and $T_2(x_2, y_2, z_2)$ at two different FoV.

We assume $T_i(x_i, y_i, z_i) = A_i \sin\left(\omega t + \underbrace{kx_i + ly_i + mz_i + \varphi}_{=\theta_i}\right)$, $i = 1,2$.

Based on the harmonic analysis, we derive $A_i$, $\omega$, and $\theta_i$. For simplicity reasons, let us assume that $y_i = 0$, $i = 1,2$:

Then, it holds $\theta_1 - \theta_2 = k(x_1 - x_2) + m(z_1 - z_2)$. As long as $z_1$ is approximately equal to $z_2$, the z-component will not influence the derived horizontal wavelength.

The other is changing the background wind needed to calculate the intrinsic frequency because the wind is averaged at airglow altitude. This error depends on the magnitude of the vertical wind shear.

The radar data are averaged over 1 h and 4 km. We took values which are ascribed to 86 km height (+/- 2 km). I understand the reviewer question in the following way: the OH-airglow altitude can be changed by tides. It can happen that it significantly disagrees from 86 km. In this case, we have used the wrong wind. This error depends on the vertical wind shear.

According to Zhao et al. (2015), strong tidal perturbations lead to airglow-altitude changes from typically 2–7 km (referring to an observation time of 10 h at maximum). When the forcing was significantly reduced or complex in nature, no systematic trends were detected.
As long as the height changes are still within the altitude bin of 86 km, I argue that we can neglect them. The situation changes, when the changes of the OH-airglow altitude are larger. Since our height bins are 4 km, we are one bin off in this case at maximum. Placke et al. (2011) show histograms of the zonal and meridional wind shear values (prevailing and tidal wind) for the years 2005–2009 referring to July measured by the radar at Collm, which we used in our publication, too. On average, the zonal (meridional) wind shear is ca. 5 m/s/km (0 m/s/km). So, if we are one bin off, this is equivalent to 20 m/s. This agrees with the error we assumed for each wind component.

If the authors do not plan to address these issues in the analysis, they should at least be discussed.
At the end of section 3.2 (error estimation) I included the following paragraph. It consists of the arguments given in this and the previous review concerning possible tidal effects.
"Errors which may arise due to tidal effects on the OH-layer height are not considered here. Strong tidal perturbations lead to airglow-altitude changes from typically 2–7 km (Zhao et al., 2005). If all four FoV are affected to the same extent (the OH-airglow altitude is shifted by a constant value), our results are not influenced. If the OH-airglow layer height increases or decreases for each FoV individually, the derived horizontal wavelengths can change. However, our results rely on spatial and temporal averages. The FoV cover 880 km² (560 km²); all values which we derive are averaged over this area. Furthermore, we analyze time series of 7 h and longer. So, effects of motions of these scales cancel out. The four FoV are rather near to each other. This reduces the possible effect of large scale motions like tides on our analysis results tremendously. Due to the redundancy of the system (we get four values for the horizontal wavelength), we dismiss results which do not agree sufficiently (as mentioned in section 3.1). This might be the case when not all but only one or two FoV are influenced by a higher or lower airglow altitude.
There is a secondary tidal effect. If the OH-airglow layer height disagrees significantly from 86 km, we use the wind information of the wrong altitude bin. Due to the width of the altitude bins, we might be one bin off in the case of strong tidal perturbations as mentioned above. The error depends on the vertical wind shear. Placke et al. (2011) show histograms of the zonal and meridional wind shear values (prevailing and tidal wind) for the years 2005–2009 referring to July measured by the radar at Collm. On average, the zonal (meridional) wind shear is ca. 5 m/s/km (0 m/s/km). So, if we are one bin off, this is equivalent to 20 m/s. This agrees with the error which we assumed for each wind component."

**About the SABER data:**
The deficiency in using SABER data to derive vertical wavelength is real and I presented the reason. The authors' arguments such as, if that's the case, then 'I can never use data …' or 'I can expand this … to every instrument' are missing the point, and are not a scientific argument. My point is, deriving vertical wavelength from a snapshot of a vertical profile has its limitation, and authors should be aware of it. This is obvious in authors' own analysis process, in which different approaches give different results, and they have to pick what's most favorable. The main message here is that this comparison can neither validate, nor invalidate the vertical wavelength derived from OH data. Consequently, whether the derived OH vertical wavelength agrees or disagrees with the SABER data is really not meaningful, because both have uncertainties due to different reasons and the uncertainties are as large as their differences. Again, it is OK for making the comparison just for reference (since there is no other data to compare), but it is not a validation of the OH derived vertical wavelength.

Thanks for clarifying this point. I have the impression that we mainly have a "wording problem" here: verification (provision of objective evidence) versus validation (consistency check of the results, see also Loew, A., et al., 2017). We would like to make a consistency check but of course we are aware of the deficits due to data coverage, the sensitivity of the instrument, etc.
Since the meaning of the term validation differs from community to community, it might not be the ideal choice in any case. In the satellite community, validation is based on the use of independent measurements of different instruments (e.g. satellite-based measurement versus radiosonde-based one). This implies that they are characterized by different deficits. In order to alleviate this point, I propose to use "compare" instead of "validate".
Therefore, I changed the beginning of paragraph 4.2 to "Vertical wavelengths derived from the scanning GRIPS are compared to vertical wavelengths extracted from TIMED-SABER temperature profiles. Therefore, we identify all nights with co-located TIMED-SABER measurements around Oberpfaffenhofen." Furthermore, I changed the last sentence of the abstract (In order to check our results, vertical temperature profiles of TIMED-SABER (Thermosphere Ionosphere Mesosphere Energetics Dynamics, Sounding of the Atmosphere using Broadband Emission Radiometry) overpasses are analysed with respect to the dominating vertical wavelength.) to "In order to compare our results …".

[revised manuscript text omitted]